

# The challenge of comparing pollen-based quantitative vegetation reconstructions with outputs from vegetation models – a European perspective

Anne Dallmeyer[1], Anneli Poska[2,3], Laurent Marquer[4], Andrea Seim[4,5], Marie-José Gaillard-Lemdahl[6]

[1] Max Planck Institute for Meteorology, Bundesstrasse 53, 20146 Hamburg, Germany

[2] Department of Geology, Tallinn University of Technology, Ehitajate tee 5, 19086 Tallinn Estonia

[3] Department of Physical Geography and Ecosystem Science, Lund University, Sölvegatan 12, 223 62 Lund, Sweden

[4] Department of Botany, University of Innsbruck, Sternwartestrasse 15, 6020 Innsbruck, Austria

[5] Chair of Forest Growth and Dendroecology, Institute of Forest Sciences, Albert-Ludwig-University of Freiburg, 79106 Freiburg, Germany

[6] Department of Biology and Environmental Science, Linnaeus University, Barlastgatan 11, 39182 Kalmar, Sweden

Correspondence to: Anne Dallmeyer (anne.dallmeyer@mpimet.mpg.de)

## Abstract

We compare Holocene tree-cover changes in Europe derived from a transient MPI-ESM1.2 simulation with high spatial resolution LPJ-GUESS time-slice simulations and pollen-based quantitative reconstructions of tree cover based on the
REVEALS model. The dynamic vegetation models and REVEALS agree with respect to the general temporal trends in tree cover for most parts of Europe, with a large tree cover during the mid-Holocene and a substantially smaller tree cover closer to the present time. However, the decrease in tree cover in REVEALS starts much earlier than in the models indicating much



earlier anthropogenic deforestation than the prescribed land-use in the models. While LPJ-GUESS generally overestimates tree cover compared to the reconstructions, MPI-ESM indicates lower percentages of tree cover than REVEALS, particularly

in Central Europe and the British Isles. A comparison of the simulated climate with chironomid-based climate reconstructions reveals that model-data mismatches in tree cover are in most cases not driven by biases in the climate. Instead, sensitivity experiments indicate that the model results strongly depend on the tuning of the models regarding natural disturbance regimes (e.g. fire and wind throw). The frequency and strength of disturbances are – like most of the parameters in the vegetation models – static and calibrated to modern conditions. However, these parameter values may not be valid

during climate and vegetation states totally different from todays. In particular, the mid-Holocene natural forests were probably more stable and less sensitive to disturbances than present day forests that are heavily altered by human interventions. Our analysis highlights the fact that such model settings are inappropriate for palaeo-simulations and complicate model-data comparisons with additional challenges. Moreover, our study suggests that land-use is the main driver of forest decline in Europe during the mid- and late-Holocene.

**1 Introduction**

Terrestrial land cover is one of the key components of the Earth's ecosystem and a provider of many ecosystem services. It is widely discussed in the context of ongoing climate change, due to its high sensitivity to environmental changes and its role as one of the mitigation agents of the current and projected global warming (e.g. Williamson, 2016; Harper et al., 2018; Smith et al., 2016). Decisions on strategies for the future depend, among others, on our ability to correctly understand the

interactions between vegetation and climate over short and millennial timescales. This requires also that Earth System Models (ESM) correctly simulate these interactions in the past to ensure reliable model projections (Harrison et al., 2020). In this context, Dynamic Global Vegetation Models (DGVM) are used, either coupled to ESMs or offline, to simulate past or future climate- and human-induced changes in land-cover composition, biomass production, and carbon storage capacity (e.g. Hickler et al., 2012; Wramneby et al., 2010; Hopcroft et al., 2017; Lu et al., 2018, 2021). However, the DGVM

parametrization (bioclimatic limits, disturbance intervals, fire regimes, etc.) are commonly static and based on the current state of land cover although it is characterized by unstable vegetation composition due to rapidly changing natural and anthropogenic stressors (Hengl et al., 2018). This is one of several caveats of DGVMs that may lead to erroneous projections for the future. Comparison of DGVM simulations for the past with proxy records of Holocene vegetation composition is a way to evaluate the performance of DGVMs. Europe is one of the most intensively studied areas of the world in terms of

number and density of pollen records and number of pollen-based reconstructions of regional plant cover at semi-continental and continental scales (Trondman et al., 2016; Marquer et al., 2017; Githumbi et al., 2022a; Dawson et al., 2018). Moreover, these reconstructions were successfully combined with auxiliary datasets (four covariates: latitude, longitude, elevation,





independent scenarios of past deforestation) to create spatially continuous maps for 6 ka (Pirzamanbein et al., 2014) and continuous time windows from 11.7 ka BP to present (Githumbi et al., 2022b) using spatial statistical models.

Pollen-based vegetation reconstructions indicate large changes in plant-species composition and distribution over the Holocene. They represent both natural and human (land use)-induced changes, the latter increasing gradually from 6 ka BP until today. These land-use related land-cover changes could potentially have a large impact on the complex climate-vegetation interactions and represent a significant climate forcing in the past (Ruddiman et al., 2015; Ruddiman, 2007; Boy et al., 2022; Huang et al., 2020). Regional climate-model simulations have shown that the anthropogenic deforestation of

Europe at 6k BP according to the KK10 scenarios (Kaplan et al., 2009) and the pollen-based land-cover reconstructions of Githumbi et al. (2022a) result in regional cooling or warming of 1 °C depending on the region and season (Strandberg et al., 2022; Strandberg et al., 2014)

Earlier evaluations of the performance of DGVMs by comparing model-simulated land cover with pollen-based plant-cover reconstructions have shown clear differences between the two for the Early Holocene and over the last 6000 years (e.g.

Marquer et al., 2017). The largest discrepancies are found in the abundance/cover of open land, with models generally underestimating the extent of unforested land. While these mismatches are commonly associated with biases in climate inputs (Strandberg et al., 2022), an increasing amount of evidence shows good conformity between climate model outputs and climate reconstructions inferred from other proxies than pollen. It implies that mismatches are rather related to the lagged reaction of trees to climate change (Dallmeyer et al., 2022) and the increasing effect of anthropogenic land-cover

change in Europe from 6000 years ago (Kleinen et al., 2011; Braconnot et al., 2019).

The aim of this study is to analyse the Holocene vegetation change in Europe with the focus on exploring the mechanisms behind discrepancies between simulated and reconstructed vegetation distributions. We compare tree cover changes simulated by two commonly used DGVMs with different inherent vegetation representation and parametrization, the MPI-ESM1.2 land-model component JSBACH and the DGVM LPJ-GUESS, with pollen-based REVEALS plant-cover

reconstructions for six time windows of the Holocene and five areas along S-N and W-E transects through central and northern Europe. Henceforth all ages are given in calibrated 14C kilo years BP, abbreviated "ka".

## 2  Methods

Figure 1 summarises the strategy for the comparison between the Dynamic Global Vegetation Models (DGVMs) JSBACH (as interactive component of the Earth System Model MPI-ESM1.2) and the offline DGVM LPJ-GUESS, and the

comparison between the DGVMs and the pollen-based plant-cover reconstructions using the REVEALS model. The different models, simulations and methods involved are described in detail below.



### 2.1 The Earth System Model MPI-ESM1.2

### 2.1.1 Model description

The model MPI-ESM1.2 (Mauritsen et al., 2019) consists of the general circulation model of the ocean MPIOM (Jungclaus
et al., 2013) coupled to the atmospheric general circulation model ECHAM6.3 (Stevens et al., 2013). MPIOM includes the global ocean biogeochemistry model HAMOCC (Ilyina et al., 2013). Vegetation and terrestrial carbon-cycle dynamics are calculated by the land-surface scheme JSBACH3 (Reick et al., 2021 and 2013), incorporated in ECHAM6.3. JSBACH3 includes the soil carbon model YASSO (Goll et al., 2015), a 5-layer hydrology scheme (Hagemann and Stacke, 2015) and the dynamic vegetation module developed by Brovkin et al. (2009). In this module, natural vegetation is represented by eight
different plant functional types (PFT). Trees can either be tropical or temperate, evergreen or deciduous. Grassy types are distinguished in C3 and C4 grass and the last two types represent raingreen and cold resistant shrubs. Furthermore, three anthropogenic land-use types are included (C3 and C4 pasture and crops). Different PFTs can coexist in each grid cell as the model uses a tiling approach, i.e. the grid cell is tiled in mosaics of fractional PFT coverages. The establishment of each natural PFT is constrained by temperature thresholds representing their respective bioclimatic tolerance. The fractional cover
of each PFT is, by and large, determined by the relative differences in annual net primary productivity (NPP) between the PFTs. Natural mortality and disturbances such as wind throw and fire reduce the cover fraction of PFTs. Woody PFTs are generally favoured at the expense of grass, but in regions with frequent disturbances or bioclimatic conditions near the bioclimatic thresholds, shrubs or even grass may win the competition as they can recover more quickly than trees. The relative presence of grasses and woody PFTs is thus implicitly determined by the strength of the disturbances. While fire has
a different effect on grass than on woody PFTs, wind throw only reduces the woody PFT types. The disturbance rate of wind throw is proportional to the simulated wind power, but it is weighted by the averaged wind speed in each grid cell to account for the adaptation of woody plants to local wind conditions. In addition, wind throw is set to zero if wind speeds are below a threshold that is determined by the long-term average maximum wind speed.

For each grid cell, JSBACH calculates the fraction of the grid-cell that is not covered by vegetation (bare soil fraction) that
represents both seasonal and permanently unvegetated ground. Since the area of bare soil cannot be estimated by the REVEALS model (see. 2.3), the PFT fractions in this study are scaled based on the total area covered by vegetation, i.e. the bare soil fraction is not considered and the cover fractions of the PFTs are adjusted to sum up to 1 in each grid cell. More details on the dynamic vegetation module can be found in Brovkin et al. (2009); Reick et al. (2021 and 2013).

### 2.1.2 Transient simulation in MPI-ESM1.2

After having been run in quasi-equilibrium for mid-Holocene (7950 BP) boundary conditions, the model MPI-ESM1.2 was used to produce a transient simulation for the period 7950 - 100 BP (Bader et al., 2020; Dallmeyer et al., 2020). The



atmosphere and land model was applied with a spectral resolution of T63 (approx. 200km on a Gaussian grid) with 47 levels in the vertical. The ocean model has been configured in the horizontal resolution GR15 (i.e. 256x220 on a bipolar grid, 12 to 180km) with 64 vertical levels. The transient simulation was performed using the following forcings:

a)   orbital-induced insolation changes (Berger, 1978), updated every decade

   b)   Methane, nitrous oxide, and carbon dioxide concentrations inferred from ice core records (F. Joos, personal communication; see Köhler, 2019 and Brovkin et al., 2019), updated every decade

   c)   stratospheric sulphate aerosol injections imitating volcanic eruptions, prescribed from the Easy Volcanic Aerosol (EVA) forcing generator (Toohey and Sigl, 2017), read annually, but calculated daily by linear interpolation

d)   Spectral Solar Irradiance forcing, includes extrapolated 11-years solar cycle based on sun-spot observations-sets of far infrared, near infrared and visible radiation (Krivova et al., 2011), read annually, but calculated daily by linear interpolation

   e)   A preliminary version of the LUH2 dataset (Hurtt et al., 2020): This forcing begins 1100 BP, but to slowly build up the land-use from zero to the first land-use state in the LUH2 dataset, a transition period of 1000 years is
implemented starting at 2100 BP. Land-use is read annually, but calculated daily by linear interpolation. Land use is prescribed in the form of transition maps that define the fraction of area that is converted from natural vegetation to crops and pasture or vice versa. Pasture is first distributed in the area covered by grass before it replaces forested area (Reick et al., 2021). After this rule has been applied, the remaining anthropogenic land-cover change is equally distributed to all PFTs, relative to their individual cover fractions, so that they experience the same gain or loss of
cover fraction.

A detailed description of the transient simulation and the forcing mechanisms can be found in Bader et al., 2020, Brovkin et al., 2019 and Dallmeyer et al., 2020. An evaluation of the simulated pre-industrial climate is provided in Appendix A.

## 2.2 The dynamic vegetation model LPJ-GUESS

### 2.2.1 Model description

LPJ-GUESS (Lund-Potsdam-Jena General Ecosystem Simulator) is an individual-based dynamic ecosystem model optimised for global to regional studies (Smith et al., 2014; Sitch et al., 2003; Smith et al., 2001). The model employs a representation of vegetation dynamics (successional processes: establishment, growth, mortality) of a forest "gap" model allowing explicit representation of competition for resources (light, water, nutrients etc.). Growth of individuals or cohorts is simulated in a number of replicate patches of 0.1 ha representing the grid cell. The climatic conditions and soil type are
assumed to be identical between the patches. The probability of stochastic patch-destroying disturbance (fire, wind etc.) occurrence is controlled by preset generic disturbance intervals. When disturbances occur, the vegetation cover of the one



patch is destroyed. The proportion of vegetation affected is dependent on the total number of prescribed patches. The vegetation is simulated as Plant Functional Types (PFT) discriminated in terms of bioclimatic limits and physiological characteristics. The standard global PFT set comprises 10 woody PFTs representing major higher plant types of boreal,

temperate and tropical biomes and two grass PFTs distinguished by C3 and C4 photosynthetic pathways. Land use is implemented using external inputs determining the proportional distribution of up to seven land-cover types (natural vegetation, urban areas, cropland, managed forest, pastureland, peatland and barren land) and associated specialised PFT set to simulate the land-cover dynamics and biogeochemical fluxes (Lindeskog et al., 2013).

The model has been applied and benchmarked in a number of studies, both for global and European conditions (Smith et al.,

2008; Hickler et al., 2012) and agricultural landscapes (Müller et al., 2021; Lindeskog et al., 2013). It is among the best available C cycle models (Piao et al., 2013) and can account for C-N interactions (Smith et al., 2014). Model performance in terms of reproducing vegetation and hydrological and biogeochemical cycles for past, present and future applications has been tested in numerous studies (Garreta et al., 2010; Miller et al., 2008; Olofsson and Hickler, 2008).

### 2.2.2 High-resolution time-slice simulations with LPJ-GUESS

LPJ-GUESS was forced with ca. 120 year monthly resolved climate (total cloud cover, precipitation and 2m air temperature) from the transient MPI-ESM1.2 simulation. Model runs were performed for six time slices of which four are distributed at nearly equal time intervals between 8ka and 2ka (representing mid and late Holocene) and two fall in the period with prescribed land use in the MPI-ESM1.2 simulation. The periods of the input climate data have been selected based on the two criteria of being close to the period of interest (e.g. 8ka) and showing a relatively stable climate, characterised by little

effects of the prescribed volcanic activity in the MPI-ESM1.2 simulation on the regional climate. The age intervals and acronyms of the six time slices are provided in Table 1.

To minimise the effect of systematic biases in the simulated climate on model-simulated vegetation, an anomaly approach was used (cf. Wohlfahrt et al., 2008). First, the anomaly between each month of each year of a simulated period (e.g. 8ka) and the respective climatological monthly mean at the end of the simulation (i.e. 199 BP - 100 BP) was calculated based on

the original T63 gridded MPI-ESM1.2 output. These anomalies were then interpolated bilinearly to a regular grid with a spatial resolution of 0.5° x 0.5° and added to a reference climate dataset, i.e. the climatological monthly mean of the years 1901-1930 taken from the CRU TS 4.0 dataset (Harris et al., 2020; University Of East Anglia Climatic Research Unit (CRU) et al., 2017). The anomaly approach has the advantage of preserving regional climatic gradients that are an imprint of e.g. a complex orography (Harrison et al., 1998) despite the relatively coarse spatial resolution used in the MPI-ESM1.2

simulation. In order to avoid negative values in precipitation and cloud cover (resulting from this calculation) in the LPJ-GUESS climate forcing data, all negative values were set to 0 for these variables.





The spin-up to reach vegetation and biogeochemical equilibrium was set to 300 years using the first 30 years of the detrended climate-data of the time window. Inputs of monthly climate variables (temperature, precipitation and cloud cover), soil texture data described in (Sitch et al., 2003) and land-use proportions derived from the JSBACH output (1ka and PI time-slice), along with a set of PFT specific parameters determining the bioclimatic niche and physiological parameters (growth form, leaf phenology, photosynthetic pathway, life history etc.) was used to set up the simulation. The number of simulated patches was set to 25 and the disturbance interval to standard 100 years for all model runs except for the disturbance sensitivity tests. These tests were also run with 25 patches while changing the disturbance interval in each run (25, 75, and 200 years). Each time a disturbance occurred, one patch (i.e. ca 4%) of the simulated vegetation was destroyed.

Yearly outputs of the computed PFT-specific leaf-area indexes (LAI) per grid cell were recorded. The spatial resolution of the LPJ-Guess runs was matching that of the climate inputs. Simulations ran uninterrupted for the whole time series. The model-produced annual record of PFT-specific LAI was averaged over the last 30 years of the modelled time slice and converted to fractional plant cover (FPC) by applying a simplified version of Lambert-Beer law (Sitch et al., 2003; Monsi and Saeki, T., 1953; Monsi, 2004; Prentice et al., 1993):

$$FPC(PFT) = \left(1.0 - exp\left(-0.5 * LAI(PFT)\right)\right) (1)$$

All woody FPC(PFT) fractions were summed to represent total tree cover (TC) of the grid cell, and the grass FPC(PFT) and land- use-related FPC(PFT) (for 1ka and PI only) were summed to represent the total open land cover (OC) fraction. To ensure comparability with pollen-based vegetation-cover estimates (i.e., assuming 100% vegetation cover), the model-based TC and OC were recalculated to sum up to 100% cover in each grid cell.

## 2.3 Quantitative pollen-based vegetation reconstructions using the REVEALS model

### 2.3.1 The REVEALS model

Pollen records consist of pollen counts from samples taken at many levels in a sediment or peat core. Pollen counts include counts for all identified pollen-morphological types (or taxa, named "pollen types" or "pollen taxa") corresponding to plant families, genera, groups of species or species. Pollen counts are then used to calculate pollen percentages for all pollen types and, if possible, pollen accumulation rates (PAR). The latter requires chronologies based on many 14C dates with high time resolution as well as sediment or peat sequences with relatively regular accumulation rates through time, i.e., no major events in sediment or peat deposition. These requirements imply that high-quality Holocene PAR records are few in comparison to pollen % records. Pollen productivity varies between plant taxa and pollen dispersal properties vary between pollen types (depending on their size and shape). These issues imply that pollen percentages (and PARs) from fossil pollen assemblages can only provide qualitative or semi-quantitative information on past vegetation changes, i.e., sporadic, or





regular presence of plant taxa, presence in more or less large quantities, increases and decreases of plant taxa. The REVEALS model (Regional Estimates of Vegetation Abundance for Large Sites; Sugita, 2007) was developed to estimate regional plant abundance using pollen % records from large lakes and corrects for the biases due to the inter-taxonomic differences in pollen productivity, dispersal, and deposition. However, REVEALS can also be applied with pollen records

from multiple small sites, although it generally results in larger standard errors (SEs) on the estimates of plant cover, as was demonstrated with model simulations (Sugita, 2007) and empirical data (Trondman et al., 2016). REVEALS has explicit assumptions (Sugita, 2007) of which the most critical ones in the context of this study are described in the Discussion section. REVEALS requires several parameters of which relative pollen productivity (RPP) and fall speed of pollen (FSP) are the most crucial. The model was first validated in southern Sweden (Hellman et al., 2008a) and later in several regions of

Europe (e.g. Soepboer et al., 2010), in northern America (Sugita et al., 2010), and in China (Wan et al., 2022). The spatial scale of a REVEALS reconstruction was estimated to ≥ 100 km x 100 km for modern vegetation in southern Sweden (Hellman et al., 2008b) and assumed to be in the same order of magnitude for European vegetation in general and through most of the Holocene. REVEALS is therefore well suited to produce pollen-based gridded reconstructions of plant cover at a spatial scale appropriate for comparison with DGVM simulations.

**2.3.2 Pollen-based REVEALS reconstructions of Holocene plant cover in Europe**

There are two gridded pollen-based REVEALS reconstructions of Holocene plant cover in Europe published so far (Githumbi et al., 2022a; Trondman et al., 2015). They were both produced at a 1º spatial scale for studies on land use as a climate forcing using climate models and DGVMs (Strandberg et al., 2014 and 2022). Trondman et al. (2015) performed reconstructions for five key time windows of the Holocene, while Githumbi et al. (2022a and 2022b) produced

reconstructions for 25 consecutive time windows between 11.7 ka and the present. Moreover, Holocene REVEALS reconstructions for 19 of the best pollen records from large lakes in Europe (Marquer et al., 2014) and for 36 1ºx 1º grid cells including these 19 pollen records and all other available pollen records around them (Marquer et al., 2017) were performed to study questions related to Holocene vegetation dynamics and plant diversity along N-S and W-E transects through Europe. All gridded REVEALS reconstructions for Europe follow the same protocol as described in Trondman et al. (2015) and used

globally by the PAGES LandCover6k working group (e.g. Li et al., 2023; Dawson et al., 2018). For the present study, we chose the REVEALS reconstructions from Marquer et al., (2017) as they were the only ones available for the entire Holocene at the time of our analysis and represented the best ones in terms of quality of the pollen records used. The major difference between Marquer et al. (2017) reconstructions and the more recent ones by Githumbi et al. (2022a and 2022b) is the dataset of RPPs and FSPs used (see Discussion section).

The REVEALS dataset of Marquer et al. (2017) covers large parts of northern and central Europe, i.e., Ireland, the British Isles, and several regions on a latitudinal transect from the Alps in the south to northernmost Norway in the north (Fig. 2).



The REVEALS estimates are based on 151 pollen records and available for 25 consecutive time windows over the last 11.7 ka as follows: 0-0.1 ka, 0.1-0.35 ka, 0.35-0.7 ka, and 500-year time-windows between 0.7 and 11.7 ka. The pollen records were selected from the European Pollen Database (Giesecke et al., 2014), the Alpine Palynological Database (University of

Bern, Switzerland), or provided by individual authors. The pollen records are from large lakes (≥ 50 ha) and small sites (lakes and bogs < 50 ha). The grid system, pollen-data handling, and REVEALS application (parameter setting etc.) follow the protocol described in Trondman et al. (2015), and the RPP and FSP dataset of Mazier et al. (2012) was used. The larger the number of pollen records (sites), the better the REVEALS reconstruction (Sugita, 2007; Trondman et al., 2016). However, given that the REVEALS model was developed for pollen records from large lakes, a single pollen record from a

large lake provides a reliable reconstruction of regional plant cover (Hellman et al., 2008; Trondman et al., 2016). The number of pollen records per grid cell varies between one and 32 in the dataset of Marquer et al. (2017). Only seven of the 36 grid cells include a single site, and in each case it is a large lake, therefore the REVEALS estimates are reliable. The REVEALS estimates for grid cells including only two small sites or one large bog (with few additional small sites) are considered as less reliable given that large bogs violate one of the assumptions of the REVEALS model and estimates based

on a few small sites will be biased towards local plant cover (see e.g. Li et al., 2020; Githumbi et al., 2022a for details). Such grid cells are few in Marquer et al. (2017) and indicated in Fig. 2. They are: one grid cell with one large bog and three small sites and one grid cell with two small lakes (both in southern Finland), and one grid cell with one small lake and one small bog (Scandinavian mountains, Sweden-Norway boundary). Moreover, there is one grid cell in Great Britain with four pollen records from small bogs only, which may bias the REVEALS reconstruction towards local plant cover on the bogs.

For each of the 36 grid cells the REVEALS model is run for each pollen record individually and the mean REVEALS estimates of plant cover (and their SEs) for the grid cell are then calculated for the 25 plant taxa (Tab. 2). The total cover of plant taxa within a grid cell is 100%. This is due to the fact, that estimating bare ground from pollen is a challenge. So far, only one attempt in northern China has been published (Sun et al., 2022). Plant-functional types (PFTs) were defined following Wolf et al. (2008). However, modifications had to be made as pollen-based plant cover provides the total cover of

each plant taxon irrespective of whether it belongs to one or several PFTs. Therefore, each plant taxon can be included in only one PFT (Table 2). The method used to calculate mean SEs for grid cells and the PFTs SEs and the delta method (Stuart and Ord, 1994), is described in Li et al. (2020).

### 2.4 Comparison between vegetation model-simulations and pollen-based REVEALS reconstructions

The differences between REVEALS and JSBACH estimates, REVEALS and LPJ-GUESS estimates and JSBACH and LPJ-

GUESS estimates have been assessed for tree cover, deciduous tree cover and conifer tree cover in each grid cell. We calculated the absolute value of the differences between two estimates for each grid cell, and we defined a scale of agreement based on this absolute value and the data distribution over the entire study region (i.e. absolute values for all grid



cells) for each time window. The first quartile and the median have been calculated. A good agreement corresponds to an absolute value of the differences between two estimates lower than the first quartile. An agreement corresponds to an

absolute value of the differences between two estimates situated between the first quartile and the median. A disagreement corresponds to an absolute value of the differences higher than the median. All results have been plotted using ArcGIS 10.6 to observe the spatial distribution of the differences between the different past vegetation reconstructions.

To evaluate the overall spatial dissimilarities between REVEALS and JSBACH and LPJ-GUESS regarding the total tree cover, deciduous tree cover and evergreen tree cover, the squared chord distance (Prentice, 1980) is calculated for each time

window. The squared chord distance is commonly used to calculate the dissimilarities between two sets of data that represent assemblages, i.e. plant composition. In this study we apply it to the ensemble of grid cells across our study region, i.e. we are studying how dissimilar the spatial grid compositions between REVEALS and the Dynamic Global Vegetation Models are for each time window.

### 3 Results

We compare the vegetation change simulated by the two models JSBACH (coupled in MPI-ESM1.2) and LPJ-GUESS (forced with MPI-ESM1.2 climate) with pollen-based REVEALS reconstructions. Please note, that land-cover changes (decreases or increases) are expressed in absolute fractions of the grid cells, e.g. an increase in cover by 20% at x ka from a cover of 50% of the grid cell at y ka implies that the cover at x ka is 70% of the grid cell.

### 3.1 European tree cover change since 8ka

The two models simulate a similar European potential natural vegetation history but with very different total tree-cover fractions over time (Fig. 3). LPJ-GUESS shows a dense tree coverage between 80-100% in large parts of Europe during mid- and Late Holocene. More open landscapes are simulated for the Mediterranean area (particularly southwestern Spain and southern Italy) and the mountainous regions of Scandinavia and the Alps. Vegetation is nearly constant until the prescribed land-use forcing sets in, substantially reducing the tree cover in western, central and eastern Europe at the 1ka and PI time-

slices.

JSBACH generally simulates a similar spatial gradient as LPJ-GUESS with fewer trees in the Mediterranean region and the Scandinavian mountains and the largest tree cover in central Europe. However, tree coverage is much lower in most parts of Europe during mid- and Late Holocene than simulated by LPJ-GUESS, mostly reaching tree cover fractions of 40-70%. Particularly obvious are the very low tree cover fractions in the coastal areas of the Atlantic Ocean and Mediterranean Sea as

well as in southern Europe, leading to remarkable differences between JSBACH and LPJ-GUESS during mid- to Late





Holocene. In contrast, JSBACH simulates a slightly higher tree cover fraction than LPJ-GUESS in parts of northern Scandinavia and the Alpine Region.

As in LPJ-GUESS, JSBACH indicates relatively constant tree cover distributions until the prescribed land-use is applied. Due to the higher potential natural tree cover fractions in western, central and eastern Europe, the land use has a stronger effect in LPJ-GUESS than in JSBACH, reducing the differences between the models with respect to the total tree cover fraction for the time-slices influenced by land use (1ka and PI).

The REVEALS-based estimates of European tree coverage for 8ka indicate dense forests in most of the represented regions with highest tree cover fractions along the Baltic Sea (Central and Boreal Europe) and in the Alpine region and a more open landscape on the British Isles. The reconstructed tree cover in Northern Scandinavia is also quite high and therefore strongly deviates from the model results.

Fig. 4 displays the mean trend in the different subregions simulated by the models and REVEALS, based on the different time-slice experiments. Since the models do not reveal much variability in the tree cover, the few simulated data points were linearly interpolated. The presented time series therefore only shows the long-term trend but cannot reflect the centennial variability.

For the Alpine region, the dynamic vegetation models and the REVEALS-based reconstructions compare well with total tree-cover estimates around 70-85% in the mean. REVEALS indicates a tree cover maximum at 6ka and a decreasing trend already starting at 4ka from about 86% at 4ka to approximately 56% at PI. In contrast, the models simulate relatively constant tree coverage for the period 8ka to 2ka followed by a sharp drop in the tree cover fraction after 2ka as a response to the prescribed land use. Tree cover is considerably reduced in all datasets, and nearly halved between 2ka and PI (from about 75% to 35%) in the models. The several millennia long mismatch in timing and intensity between the models and the reconstructions indicate not only a later onset of but also a too extensive deforestation in the models compared to REVEALS estimates.

For Central Europe, LPJ-GUESS simulates constant high tree coverage of more than 90% between 8ka and 2ka, and a sharp reduction (to 27% at PI) thereafter. JSBACH reveals a similar, albeit weaker, vegetation dynamic with mid-Holocene tree cover fractions of approx. 50% and a substantially decreased tree cover at PI (~22%). Also, for this region, the REVEALS-based reconstructions show a different temporal trend. The decrease in tree cover begins as early as 6ka and accelerates towards PI. At 8ka, tree cover is estimated at 85%, agreeing well with the LPJ-GUESS estimates. Tree cover is nearly halved to 41% at PI according to REVEALS. The reduction in tree coverage after 2ka parallels the trend in the JSBACH simulation, but on higher tree cover fractions.





LPJ-GUESS and REVEALS indicate similarly high (~85-90%) mid-Holocene tree-cover fractions in Boreal Europe, whereas JSBACH simulates much less trees (~60%). Both, REVEALS and LPJ-GUESS, show relatively constant tree cover between 8ka and 4ka. In REVEALS, the tree cover fraction decreases slightly from 4ka but is substantially reduced (by 36%) only in the last ~2000 years. In LPJ-GUESS, tree coverage declines by only 15% between 2ka and PI. It is difficult to figure out the effect of land use in this model-setup, but the differences in the magnitude of the tree-cover decrease could

indicate an underestimation of the prescribed land use intensity in this region in the models. In JSBACH, land use has hardly any effect at all in the regional mean.

For Northern Scandinavia, the models and the REVEALS reconstructions show a similar trend of steadily decreasing tree cover during the Holocene but differ significantly in absolute tree cover. REVEALS estimates a tree coverage of approx. 80% at 8ka and declining to 60% at PI. LPJ-GUESS simulates a mean tree cover of 58% at 8ka and 26% at PI. In JSBACH,

the simulated tree cover fraction is reduced from 40% at 8ka to 21% at PI. The relative decrease in tree coverage (tree cover is halved in both models) is thus much stronger in the models than estimated by the REVEALS-based reconstructions.

The models and reconstructions show the largest deviations on the British Isles. While JSBACH indicates very low tree cover fractions of approx. 20% at 8ka and a slight decrease in tree cover after 2ka to 13% at PI, LPJ-GUESS simulates high tree-cover fractions during the mid-Holocene (~92%) and a sharp drop to 58% between 2ka and 1ka. REVEALS estimates a

tree cover of 60% at 8ka and a constant decrease between 6ka and 2ka to 27%, followed by a stronger drop in tree cover towards 1ka (to 10%) and a slight recovery towards PI (to 14%).

In summary, inter-model spread is largest in the British Isles and Central Europe as well as in southern Europe (no REVEALS region) with much lower tree cover fractions in JSBACH than simulated by LPJ-GUESS. Compared to the REVEALS estimates, JSBACH underestimates the tree cover in practically all regions and during most of the time slices

except for the Alpine region, whereas the LPJ-GUESS estimated tree cover is well comparable to the REVEALS estimates in Boreal and Central Europe. However, LPJ-GUESS significantly underestimates the tree cover in Northern Scandinavia and overestimates it in the British Isles. The models suggest rather constant European tree coverage between 8ka and 2ka, whereas the REVEALS-based reconstructions show a stronger dynamic with the tendency of a mid-Holocene (6ka) maximum tree cover followed by a steady decline. The prescribed land use has a larger effect in LPJ-GUESS than in

JSBACH, indicating differences in its implementation and a higher tree cover level at the onset of land use in LPJ-GUESS. Compared to the REVEALS reconstructions, the prescribed land use in the models appears to be too large in parts of Central and western Europe (particularly in the Alpine region) and too small in Boreal Europe.



### 3.2 REVEALS versus vegetation models

#### 3.2.1 Temporal distribution of the differences

The calculated chord distances between the datasets indicate that the simulated tree-cover pattern in LPJ-GUESS is overall in better agreement with REVEALS during the mid-Holocene than the one inferred by JSBACH (Fig. 5). Particularly for the 8ka time slice, the total tree cover simulated by LPJ-GUESS compares very well with the reconstructions, yielding a chord distance below 1. The chord distance for the evergreen and deciduous tree cover is substantially higher, indicating that the ratio of deciduous to evergreen forest is not quite as well represented by LPJ-GUESS.

The results for JSBACH suffer from the severe underestimation of deciduous trees compared to REVEALS. This is reflected in the very high chord distance at 8ka. However, the spatial distribution of evergreen trees is better represented by JSBACH at 8ka. The agreement of the two different models to REVEALS converges towards 1ka and PI. The representation of the deciduous trees gets continuously better in JSBACH, while it gets worse in LPJ from 6ka to 2ka. This affects the model-data agreement with respect to the total tree cover. The mismatch in tree cover between LPJ-GUESS and REVEALS increases
towards 1ka. For JSBACH, the chord distance to the REVEALS estimated tree cover distribution is relatively constant until 4ka and getting better afterwards.

Since the mismatch to the reconstructions in the distribution of evergreen trees is relatively constant in both models, the improvement of the agreement in total tree cover between the DGVMs and REVEALS is clearly driven by the increasingly better representation of the deciduous trees in the DGVMs with time. Even if generally, the cover fractions of the evergreen
trees in both models agree better with the REVEALS estimates than the deciduous tree cover fractions. Since 4ka, the total tree cover distribution simulated by JSBACH agrees better with REVEALS than the LPJ-GUESS simulated distribution, but the misrepresentation of the ratio of evergreen to deciduous trees remains.

Overall LPJ-GUESS shows the best agreement with REVEALS for the 6ka time-slice, followed by the PI time-slice. JSBACH also indicates an improvement of the agreement with respect to the cover fraction of evergreen trees for 6ka.
However, it agrees best with REVEALS for PI conditions.

#### 3.2.2 Spatial distribution of the differences

We calculate a three-scale agreement index to quantify and evaluate the spatial difference between the models and the REVEALS-based reconstructions in each grid-cell (cf. Methods). The pattern indicates model-specific regions with systematic agreement or disagreement (Fig. 6). For instance, JSBACH fails to reproduce the REVEALS mid-Holocene tree
cover fraction in large parts of central Europe, the Baltic States and southern Sweden, while LPJ-Guess shows reasonably good overall agreement in these regions. Both models produce tree cover estimates not comparable with REVEALS





reconstructions for the British Isles. For JSBACH this mismatch gets slightly better towards PI, mainly due to an improved representation in some grid-cells in northern Germany, Scotland and Ireland. In most grid cells on the British Isles and in western Germany, the fractional coverage of evergreen trees simulated by both models has better agreement with the

REVEALS estimates than the deciduous tree cover fraction, underlining the previous finding of the deciduous trees as driver of the model-data mismatch. However, in eastern Europe (Poland, Baltic States) the deciduous tree coverage simulated by both models is more in line with the REVEALS estimates than the evergreen tree coverage.

The tree cover fraction in the Alpine region simulated by JSBACH corresponds better to the REVEALS estimates than in other regions until the land use in the model sets in. All grid-cells in these regions receive agreement or even good agreement

with respect to the JSBACH-derived total tree cover. In contrast, LPJ-GUESS indicates only some agreement in the central Alpine region. The fraction of deciduous trees is mostly in agreement for both models during the mid-Holocene, but the agreement decreases towards the late-Holocene. The simulated fractions of evergreen trees disagree with the REVEALS-based estimates at all time-slices.

In most grid cells of Northern Scandinavia, the deciduous and total tree cover do not match the reconstructions during all

time-slices. The evergreen trees, however, show agreement in many grid cells, which becomes even better towards 1ka.

The total tree coverage in REVEALS and JSBACH compares well in the domain southern/central Norway to central Sweden and southern Finland, reaching values of agreement to good agreement in most grid cells during all periods. LPJ-GUESS does not agree as well as JSBACH with REVEALS, particularly in southern Norway. Interestingly, in these regions JSBACH is not able to capture the REVEALS-estimated fractions of evergreen and deciduous trees, indicating a

misrepresentation of the ratio of the different tree types. In contrast, LPJ-GUESS captures the REVEALS deciduous tree-cover fraction for the Late Holocene.

Based on the results presented in Fig. 5 and Fig .6, it is not possible to determine which model is overall more consistent with REVEALS. The agreement strongly depends on the region. It should be noted, however, that in a statistical sense the thresholds for the three scale-agreement indices are equal for both models, but the absolute values of the thresholds may

differ, depending on the general spatial variability in the models that is higher in JSBACH than LPJ-GUESS. This allows JSBACH to be still rated in the category "agreement" with larger absolute differences to REVEALS.

## 4 Discussion

Comparison of the tree-cover absolute values and their spatial and temporal distributions as simulated by the Dynamic Global Vegetation Models and estimated from pollen data by the REVEALS model is challenging. When compared with

REVEALS, JSBACH rather underestimates European tree cover during mid- and Late Holocene, while LPJ-GUESS mainly



overestimates the tree coverage, at least in those time-slices without human impact prescribed to the models (8ka to 2ka). In addition, the model agreement to the REVEALS results shows a spatially varying pattern that is different for each model. Whereas JSBACH reveals the strongest mismatch to the reconstructions in Central Europe during all periods, LPJ-GUESS shows relatively good agreement in this region. Both models fail to reproduce the tree cover history on the British Isles and the in Northern Scandinavia. For the Alpine region, the simulated JSBACH tree cover fractions correspond well to the REVEALS estimates, while the LPJ-GUESS values more often disagree.

In most regions, the models are not able to simulate the correct ratio of deciduous to evergreen trees, but the evergreen tree cover distribution is generally more in line with the REVEALS data than the deciduous tree coverage. While LPJ-GUESS shows a relatively good overall agreement with REVEALS at 8ka and particularly at 6ka, the mismatch increases from 4ka to 1ka. In contrast, the distributions simulated by JSBACH continuously improve (in the mean) towards PI, indicating a convergence of the model results through time.

We assume the following possible reasons behind the model-data differences and discuss them thoroughly in the next sections:

a) climate and spatial resolution biases in the models

b) oversimplified vegetation dynamics in the models

c) modern parametrizations and tuning of the model to modern conditions

d) differences between the pollen-based REVEALS estimate of deforestation due to land use and the prescribed land use as well as differences in the land-use implementations in the models

e) shortcomings of the REVEALS model and pollen-based reconstructions of plant cover.

## 4.1 Effect of biases in the simulated climate and climate input fields on the tree cover distribution

To infer possible biases in the simulated climate trend, the anomalies of the MPI-ESM1.2-simulated temperature of the warmest month (Twarm) to pre-industrial climate are added to the CRU TS4.0 reference state and then compared to chironomid-based Twarm reconstructions (Fig. 7), extracted from the synthesis of Kaufman et al. (2020) and few other sources (references are given in Table B1 in the Appendix B). Thus, we compare basically the climate that is prescribed to LPJ-GUESS with proxy-based reconstructions. We assume that summer temperature is the main climatic driver of the vegetation in the regions considered here, as previous analysis based on a slightly different Holocene MPI-ESM1.2 simulation has revealed for most parts of the regions considered in this study (Dallmeyer et al., 2021). It should be stressed however that temperature reconstructions based on chironomids may be subject to various caveats, whose detailed description is beyond the scope of this paper. Other environmental changes in e.g., nutrient, anoxia, and salinity, can indeed also lead to changes in chironomid assemblages, which may affect the chironomid-inferred temperatures (e.g. Velle et al.,



2010). Nevertheless, the distribution of chironomid assemblages generally strongly correlates with the warm-season air and lake temperature in the temperate and subarctic region, although the causal relationships are still not fully understood (e.g. Eggermont and Heiri, 2012).

The model-based mean temperature dynamics are well in line with the reconstructions. Simulated Twarm levels in the
Alpine region are different for all sites and cover a wider range than the reconstructions indicate. While Twarm decreases relatively uniformly at all sites after 8ka in the model, two of the four reconstructions show a climatic optimum between 7ka and 4ka. This is consistent with the maximum tree-cover fraction in the REVEALS data at the 6ka and 4ka time slices in this region. The good agreement of the simulated and reconstructed mean temperature of the warmest month reflects the similarly high tree cover fractions estimated by REVEALS and simulated by both models.

For Central Europe, only one chironomid-based reconstruction exists in Kaufman et al. (2020), i.e. for Lake Zabieniec (Płóciennik et al., 2011). For this record, new reconstructions have been published recently (Luoto et al., 2019; Kotrys et al., 2020). With a mean Twarm of around 16 °C, these reconstructions indicate a substantially, roughly 5°C, cooler summer climate than prescribed to LPJ-GUESS at 8ka. Furthermore, the reconstructions reveal stable or slightly increasing Twarm for the mid-Holocene in contrast to the decreasing trend in the model. For the Late Holocene, the reconstructions of the
Zabieniec record diverge strongly, revealing contradictory changes in Twarm. Also, for the Late Holocene, all reconstructions from Zabieniec indicate a much colder climate in this region than revealed by the simulated climate that has been used to force LPJ-GUESS. However, a chironomid-based reconstruction from northern Poland (Lake Spore, Pleskot et al., 2022) is well in line with the model. Both show declining Twarm from about 19°C to 17.5°C during the Late Holocene and only reveal substantial differences for the last 500 years BP. The difference in Twarm level at Lake Zabieniec may act as
an indicator for a slightly too warm model derived climate in the southern part of Central Europe. This warm bias may also extend to the southeastern part of the simulation region, displayed exemplarily for the 4ka time slice (Fig. 7). This warm bias may contribute to the slightly overestimated tree cover fraction in LPJ-GUESS compared to the REVEALS estimates in Central Europe.

In Boreal Europe, model-based Twarm is higher by approximately 2-3°C during the entire period compared to the
chironomid-based reconstructions. While the model indicates an almost linear decrease in Twarm since 8ka, some reconstructions reveal a warming towards 6ka and decreasing temperatures afterwards. This dynamic is also visible in the mean over the region and fits nicely to the REVEALS-based reconstruction of tree cover change. Regardless of these discrepancies, the modelled temperature range is still located well within the tolerance limits of boreal tree taxa. Since the tree cover fractions are similar in LPJ-GUESS and REVEALS, this bias in temperature does not affect the total tree cover
distribution.





Simulated temperatures of the warmest month are on a similar level as the reconstructions in Northern Scandinavia during mid-Holocene, not explaining the 20% lower tree cover fraction in LPJ-GUESS compared to the REVEALS estimates. The reconstructions reveal only a slight decrease in Twarm since 8ka, in contrast to the stronger decrease in the model. These differences in temperature trend are in line with the slight deviations in the rate of tree cover decline between LPJ-GUESS

and REVEALS in Northern Scandinavia. However, both modelled and reconstructed Twarm averages fluctuate close to the 10 °C, with the modelled temperature falling below it during the Late Holocene. The 10°C limit is - according to the known Köppen's Rule (Köppen, W., 1936) - accepted as delimiter of boreal forest distribution, and Twarm falling under this limit could be a possible cause for low model-based tree cover estimates.

Tree-cover fractions simulated by LPJ-GUESS show the highest discrepancy with the REVEALS estimates on the British

Isles, overestimating tree coverage by more than 30%. While the mean trend in Twarm during the period 6ka to 3ka is relatively similar (slightly decreasing) between the prescribed climate and the chironomid-based reconstructions, LPJ-GUESS shows a constant, large tree cover until land use sets in and REVEALS indicates a strong decrease in tree cover starting at 6ka. These deviations in total tree-cover changes through time cannot be explained by the simulated summer temperature.

Comparing the chironomid-based reconstructions with the climate simulated by MPI-ESM1.2 directly is not very meaningful, since many of the chironomid sites are located in mountainous regions. Due to the coarse spatial resolution of the model, one can expect the climate to be rather too warm and to show much stronger differences in annual extremes than in the mean over the seasons. The evaluation of the pre-industrial climate (Appendix A and Fig. A1 in the Appendix A) reveals only minor differences in the seasonal temperature mean between the CRU TS4.0 data and the simulated MPI-

ESM1.2 climate for PI. The total tree coverage simulated by JSBACH agrees well with the LPJ-GUESS results in the Alpine region and Northern Scandinavia with Twarm limited tree growth. JSBACH strongly underestimates tree coverage in Central Europe and for the British Isles. The former region does not experience any substantial differences in seasonal temperature, and the wetter climate in MPI-ESM1.2 probably does not induce changes in total tree coverage as the land cover of the region is not moisture-limited. The latter region is affected by strong deviations in the precipitation pattern with a much drier

and rather warmer climate at most grid-cells for which REVEALS reconstructions are available. Since the strong differences in tree coverage compared to LPJ-GUESS cover the entire British Isles, and thus also the regions with overestimated precipitation, these climate biases cannot be responsible for the strong deviations in vegetation composition between the models.

In Boreal Europe, winter and summer climate in MPI-ESM1.2 is slightly cooler than observed. Since the climate is

nevertheless well within the climatic tolerance range for extratropical tree PFTs in JSBACH, this does not limit the establishment of trees in JSBACH. However, while JSBACH has only one deciduous and one evergreen PFT for temperate



and boreal conditions, the PFT list of LPJ-GUESS includes several PFTs developed specifically considering boreal conditions (Table 2). This may contribute to the substantially lower tree coverage in JSBACH compared to LPJ-GUESS.

We conclude that climate biases are not the main driver of differences in the total tree-cover fraction between the REVEALS estimates and the vegetation models and between LPJ-GUESS and JSBACH in most of the regions. The simulated climate trends are mostly in line with the chironomid-based reconstructions and in case of differences, these cannot cause the discrepancies in the tree-cover trend. Therefore, it is highly likely that land use is the reason for the much earlier tree-cover decline in the reconstructions than in the models and that land use is the main driver of the long-term Holocene tree-cover change in Europe.

**4.2 Effect of oversimplified vegetation and soil dynamics in the models**

Even though we have adjusted the PFT distributions calculated with the different methods to be more compatible with each other, there are various technical reasons that can lead to differences between the Dynamic Global Vegetation Models and between these and REVEALS. Many processes can only be represented in a simplified form or have not been implemented in vegetation models yet. For instance, vegetation is only aggregated in a few PFTs, which neither match in number or

definition to each other nor to REVEALS (Table 2) and not all PFTs considered in REVEALS have a counterpart in the models. Whereas REVEALS reconstructions reflect actual land cover including understorey vegetation, wetlands and other specialised communities, the models estimate terrestrial high ground vegetation only. In contrast, the models calculate a fraction of uncovered soils that cannot be determined in pollen-based reconstructions and is therefore not included in the REVEALS data. LPJ-GUESS considers early and late successional deciduous and evergreen tree types separately, whereas

JSBACH distinguishes only one single deciduous and evergreen tree type, respectively. This differentiation allows for quick re-establishment of the tree cover after disturbance or climate induced mortality, making LPJ-GUESS simulated forest cover considerably more stable and less affected by random deforestation events.

Seed dispersal is not included in either of the models. Seeds are assumed to be available everywhere all of the time, whereas in the real world they need to be transported before a tree can grow in a new spot under favourable environmental and site

conditions for tree growth. Moreover, depending on existing vegetation and tree species, the success of the establishment of a tree species might differ and thus their regional abundance. Although the dispersal- and migration-related delays in establishment can mostly be expected in connection with the reforestation of Europe during the post-glacial and early Holocene period (Giesecke et al., 2017), they could be the reason for the differences in the mid-Holocene tree-cover changes between the models and REVEALS. The models show rather stable, in some regions slightly decreasing, mean tree cover

with time, while REVEALS estimates tree-cover maxima between 6ka and 4ka for grid cells in Boreal and Central Europe and in the Alpine regions.





In addition, the models have simplified soil dynamics and do not consider changes in soil type or soil build-up. Permafrost soils, peat- or wetlands and blanket bogs are not represented in these simulations. Therefore, the models lack representation of important habitats such as mires and bog, whose (mainly treeless) vegetation increases the openness in reconstructed
vegetation. Most of Europe had substantially more wetlands in the past than at present with its highly drained landscapes (Čížková et al., 2013). Furthermore, several European countries have ca 15 (Sweden, Ireland and Scotland) - 30% (Finland) of territory occupied by blanket bogs and wetlands even today. Fyfe et al. (2013) highlight the considerable bias in landscape openness between the British Isles and continental Europe throughout the Holocene in the REVEALS-based study and suggest that this could at least partly be due to the considerably higher proportion of wetlands and uplands in the land cover
of the British Isles compared to continental Europe. The formation of blanket bogs, typical for cool and hyper oceanic climates, was accelerated by climate cooling starting ca. 6 ka (Gallego-Sala et al., 2016), which may, together with anthropogenic deforestation, explain the small woodland cover. As a result of the disregard of wetlands, models tend to overestimate the tree-cover fraction in wetland-rich regions. This may explain the deviations between REVEALS and LPJ-GUESS revealing much higher tree cover fractions in the British Isles and slightly higher tree cover in Boreal and Central
Europe than REVEALS.

### 4.3 Effect of modern parametrizations and of tuning of the model to modern conditions

Each vegetation model uses a different way of representing vegetation or incorporating processes such as PFT establishment, plant competition, natural mortality, or reductions of plants by disturbances such as fire, wind throw and insects. However, they share the implemented equations of these processes and the thresholds used (e.g. for the bioclimatic tolerance) are
validated and calibrated using modern observations. Parameters are tuned to meet modern vegetation distributions and do not change in time. This means that all basic settings such as bioclimatic limits, allocation and mortality timescales or sensitivity to disturbances are assumed to be constant over the simulation time. While it is true that species-specific bioclimatic limits change slowly and most of the simulations do not reach the timescales necessary for considerable changes, the validity of the assumption that present-day plant distribution is in equilibrium with climate is questionable. Modern species-distribution
patterns are heavily influenced by centuries of agriculture and forest management. The recent, abrupt global climate change combined with other human impacts on plant species has enhanced the difference between potential and realised niche, making it highly doubtful that the current species-distribution limits would represent the actual bioclimatic envelope of plant species in Europe. Furthermore, these values may not be valid for climate states totally different from today, such as the mid-Holocene; this also refers to the concept of no-analog communities (e.g. Williams and Jackson, 2007). The simulated tree
cover in areas occupied by forest biomes is largely dependent on the implemented disturbance extent, severity and interval. However, the causes and consequences of disturbances are largely different in natural- and human-influenced landscapes. The mid-Holocene natural forests were probably much more stable and less sensitive to disturbances than present-day forests





that are heavily altered by human interventions. We hypothesize that the large underestimation of the tree fraction on the British Isles and Central Europe in JSBACH compared to REVEALS is a consequence of too much wind throw in the model. In contrast, the overestimated tree-cover fraction in LPJ-GUESS may at least partly be related to a too small spatial scale and frequency of disturbance occurrences in LPJ-GUESS. We tested this hypothesis in a sensitivity study in which we extended the MPI-ESM1.2 spin-up run for 8ka with halved sensitivity of the trees to the wind throw (i.e. halving the wind damage scaling parameter in JSBACH). For LPJ-GUESS, we performed a set of simulations changing the interval between disturbance occurrences from 25 years to 50 years, 75 years and 200 years using the 8ka climate forcing. The standard setup used in the comparison with REVEALS and JSBACH above is 100 years.

Wind throw reduction substantially increases the tree cover simulated by JSBACH in a broad area in mid-Europe (48-58°N), including the regions of the British Isles and Central Europe, and along the Norwegian Atlantic coast (Fig. 8). These are also the regions in which JSBACH substantially deviates from REVEALS and strongly underestimates tree cover in the transient simulation. However, in the other regions, only few REVEALS grid cells are affected, thus in regional means over the grid cells, the effect of wind throw is smaller.

The disturbance frequency tests with LPJ-GUESS show that each reduction of the occurrence interval by 25 years leads to approximately 4% less tree cover. This reduction is mostly related to a decrease in the cover of the evergreen PFTs (Fig. 8). The deciduous PFTs are not so heavily affected, as the disturbance gives some advantage to the early-successional deciduous PFT (Table 2), parametrized keeping in mind quick establishment and growth. Therefore, the shortened disturbance interval leads to an increased representation of deciduous trees at the expense of evergreen ones in northern and eastern Europe (Fig. 8). While LPJ-GUESS-simulated total tree cover is in general already in rather good accordance with REVEALS reconstructions, especially during mid-Holocene, the conformity could be improved by increasing the disturbance frequency.

## 4.4 Effect of land use

European land cover has been affected by humans during most of the Holocene. Pollen-based studies show first traces of crop cultivation in southern Europe more than 10000 years ago an its spread to the southern fringe of northern Europe during the following six millennia (Githumbi et al., 2022a). This transition to agrarian subsistence led to substantial anthropogenic deforestation of large parts of Europe. Particularly early, the coastal areas of western Europe were strongly affected and had already lost half of the natural forest cover 3500-5500 years ago (Roberts et al., 2018). While pollen-based studies can give detailed insights of land-cover development in an area, these records are often not quantitative, spatially discrete and do not have high (preferably annual) resolution, all prerequisites of input datasets for vegetation models. New, pollen-based reconstructions by Githumbi et al., 2022b address most of the above-listed obstacles, providing excellent quality, quantitative and spatially discrete proxy-based continental scale land cover reconstructions for Europe during the Holocene.





However, while these reconstructions are suitable for usage as land-cover representation in climate models or for validation of vegetation model performance, reconstructions do not distinguish between the natural and anthropogenic land-cover types

and therefore cannot be directly used as an anthropogenic land cover change (ALCC) input to vegetation models. There are no attempts at using land use inferred from pollen-based REVEALS reconstructions of plant cover in DGVMs published so far. Most DGVMs use prescribed land use derived from various ALCC scenarios such as KK10 (Kaplan et al., 2009) and HYDE 3.2 (Klein Goldewijk et al., 2017). However, KK10, HYDE and other ALCCs exhibit large discrepancies in their estimates of the starting time, spatial pattern and intensity of anthropogenic land-cover change, making it a challenge to

simulate human-induced vegetation with DGVMs (Kaplan et al., 2017; Gaillard et al., 2010). Here we used a preliminary version of the LUH2 dataset by (Hurtt et al., 2020), that assumes no human interference with land cover prior to 2 ka. The increased disagreement between LPJ-GUESS and REVEALS for the British Isles and some Central and Boreal European sites for 4 ka and 2 ka is probably due to considerable anthropogenic deforestation of these areas already prior to 2ka. During the last two millennia the overall agreement between the models (especially JSBACH) and REVEALS increases, showing

the significance of accounting for anthropogenic deforestation.

In the models, substantially different approaches are used to handle land use. While in LPJ-GUESS, a certain grid-cell fraction is reserved for land use related land-cover types, JSBACH calculates natural vegetation first and afterwards applies land transitions with land-use types preferentially replacing grasslands. These differences in implementation of land use explain the larger impact of land use on LPJ-GUESS-simulated than JSBACH-simulated tree cover. LPJ-GUESS-simulated

natural vegetation dynamics agree well with REVEALS-estimated plant cover at 8ka and 6ka, while JSBACH underestimates tree cover. The more the landscape is affected by humans, the greater the differences in tree cover between LPJ-GUESS and REVEALS, since the full scope of actual land use in the past cannot be reproduced by the models. JSBACH simulates a rather open landscape and prescribed ALCC has - due to specifics of its implementation - relatively little impact on the tree cover fraction. The combination of these two characteristics leads to a convergence of REVEALS

and JSBACH tree-cover over time and space through the last two millennia. LPJ-GUESS, differently from JSBACH, applies the prescribed ALCC proportions directly, making the accuracy of the used ALCC dataset especially important. The increased disagreement of the models compared to REVEALS in the Alpine region suggests that a too strong anthropogenic deforestation has been prescribed in this area.

**4.5 Caveats of the REVEALS model and pollen-based reconstructions of plant cover**

Many of the assumptions of the REVEALS model are violated in the "real world" and/or violated in the past, which has been described and discussed in detail earlier (e.g., Hellman et al., 2008a; Sugita et al., 2010; Mazier et al., 2012; Li et al., 2020). Major assumptions are: a) there is no vegetation growing on the basin (i.e., REVEALS was developed for pollen records from lakes), b) wind comes from all directions and wind speed is constant through time, and c) pollen productivity of plant



taxa and fall speed of pollen are constant through time. Moreover, the REVEALS model is not well suited for pollen-based

reconstructions in mountainous areas, which is discussed in Marquer et al. (2020). Given that the model does not account for topography, a flat topography can be considered as a model assumption, although not mentioned as such in Sugita (2007). These assumptions are violated to various degrees. Bogs are covered by vegetation, and therefore violate a major assumption of the REVEALS model. Pollen records from bogs are used in the continental REVEALS reconstructions (e.g., Githumbi et al., 2022a) and in the dataset of Marquer et al. (2017) that have been used in this study. The decision to use these pollen

records is motivated by a) using only pollen records from lakes would decrease significantly the number of available pollen data, and b) Trondman et al., (2016) showed that pollen records from small bogs can be used in REVEALS reconstructions based on multiple pollen records, given that these are used together with several pollen records from small and large lakes. We know that wind speed and direction do influence pollen assemblages in lake sediments or soil samples (e.g. Nielsen, 2003) and might therefore influence REVEALS reconstructions. We also know that wind speed and direction changed

through the Holocene (e.g., Björckl and Clemmensen, 2004; de Jong et al., 2007 and 2006; Nielsen et al., 2016), but such studies are too few to be considered in REVEALS reconstructions for a whole continent. So far, REVEALS reconstructions of plant cover assume a constant wind speed through space and time, 3 m/s for Europe. Pollen productivity is assumed to be a taxon-specific constant through the Holocene. The within-taxon variations observed in estimated relative pollen productivities (RPPs) based on modern pollen-vegetation datasets collected in the field are assumed to be due to a

combination of between-study differences in methodologies, climate, vegetation types and land use (Broström et al., 2008). There are still too few RPP values to perform meaningful statistical analyses of the possible effects of these factors, and therefore are still not considered for the past. Thus, all existing syntheses of RPP values published so far advise to use mean taxon-based RPP values for REVEALS reconstructions, in this case mean RPPs based on all available values over Europe (Broström et al., 2008; Mazier et al., 2012; Wieczorek and Herzschuh, 2020; Githumbi et al., 2022a).

The effects of the violation of assumptions cannot be quantified and accounted for, i.e., REVEALS estimates cannot be corrected for these effects. However, in the context of this study, they can be considered as possible causes behind discrepancies between REVEALS estimates of plant/PFT cover and DGVMs simulated PFT cover. The fact that topography is not considered in REVEALS implies that REVEALS estimates in the Scandinavian mountains and the Alps are uncertain and may explain discrepancies with DGVMs at the grid-cell scale level (Fig.3). At the regional scale, REVEALS, LPJ-

GUESS and JSBACH agree when standard deviations are considered (Fig. 4). REVEALS tree cover for the grid cell with four pollen records from small bogs (Britain; Fig.2) may be biased towards local plant cover on the bogs (i.e., overestimated cover of open land) because there are no additional pollen records from large lakes (or several small lakes) in the grid cell that would "correct" the mean REVEALS estimate towards less open land. This could contribute to the significant discrepancy between REVEALS and the two DGVMs for this grid cell.



Beside the violation of model assumptions, there are other features of REVEALS reconstructions that may play a role in the REVEALS-DGVM comparison. REVEALS reconstructs the cover of individual plant taxa rather than the cover of plant functional types or land-cover types (such as "open land"). When the REVEALS taxon-based estimates are summed into PFTs or land-cover types, the cover of individual plant taxa cannot be distributed between several PFTs or land-cover types. This should not affect the comparison with the PFT cover simulated by both models in this study, given that the simulated

woody PFTs include only woody plants, and open land only herbs. Acceleration of the development of *Calluna* heathland at the expense of woodland from 6 ka on due to land use (grazing and burning) in coastal areas of westernmost Europe (Nielsen et al., 2012) could explain some discrepancies in the cover of open land between REVEALS and the DGVMs, however at the grid-cell scale level only (Fig. 3). The REVEALS cover of *Calluna* is indeed between 10 and 60 % from 6ka to 1ka in several grid cells of the British Isles, Denmark and southern Norway (Trondman et al., 2015; Marquer et al., 2019).

Another feature of the REVEALS model is the use of pollen data implying that the model reconstructs the cover of plants that produce pollen. This is mainly critical in the case of trees and shrubs that may start to produce pollen after many years. Therefore, the cover of young trees is not included in REVEALS tree cover that may therefore underestimate tree cover. However, REVEALS tree cover is never significantly lower than the tree cover simulated by the DGVMs, except for the British Isles where tree cover is larger for LPJ-GUESS than for both JSBACH and REVEALS in all time windows except 1

ka and PI. Finally, the REVEALS reconstructions used in this study are based on 25 plant taxa only. However, the pollen types ascribed to these plant taxa represent > 90% of the pollen counts and most missing taxa belong to herbs that would not decrease the REVEALS tree cover very significantly.

In summary, there are few caveats of the REVEALS model itself and the REVEALS dataset used in this study that can contribute to the discrepancies between REVEALS-estimated and DGVM-simulated tree cover. The discrepancy between

REVEALS and LPJ-GUESS in the British Isles is obviously due to the significant land-use in this region from 6 ka on and the lack of wetlands in the model. The mismatch of JSBACH in Central Europe, Boreal Europe, and the British Isles, i.e. the underestimation of forest cover in comparison to LPJ-GUESS and REVEALS, indicate that the implementation scheme (and the tuning) of JSBACH is problematic for these regions in particular.

### 5 Summary and Conclusions

We compare pollen-based quantitative reconstructions of Holocene tree cover in Europe estimated by REVEALS with a transient simulation of the last 8000 years undertaken with the Earth System Model MPI-ESM1.2 (including the dynamic vegetation model JSBACH) and time-slice simulations conducted with the dynamic vegetation model LPJ-GUESS. Both models and the reconstructions indicate larger tree cover in most parts of Europe at 8ka compared to PI but differ substantially with respect to the total area covered by trees and the age of the start of deforestation. While LPJ-GUESS





generally overestimates tree cover fractions compared to REVEALS, JSBACH indicates much lower percentages of forested

area in most parts of the region, albeit with a similar spatial pattern as LPJ-GUESS.

The total area covered by trees is relatively constant in the models until the prescribed land-use sets in, i.e. after the 2ka time

slice. In contrast, REVEALS indicates a 6ka maximum in tree cover in some grid-cells in Central and Boreal Europe and

particularly in the Alpine region. A comparison of the simulated climate with chironomid-based climate reconstructions

reveals that climate biases only marginally cause these disagreements between the simulated and reconstructed trend in tree

cover. Instead, the reconstructed 6ka maximum in some areas may be related to dispersal and migration-induced delays in

the establishment of some tree taxa. These processes are not included in the models.

According to REVEALS, anthropogenic deforestation starts much earlier (~4ka in the Alpine region and 6ka in Central

Europe and the British Isles) than in the model forcing. While the decline in the tree cover fraction in REVEALS is relatively

steady, the prescribed land use induces a sharp drop in tree cover in most regions in the models, indicating a too intensive

land use in central and western Europe, particularly in the Alpine region. Prescribed land use in Boreal Europe seems to be

too weak in the models compared to the REVEALS estimates. The prescribed land use has a larger effect in LPJ-GUESS

than in JSBACH, pointing to differences in the implementation of land-use and the higher tree cover level at the onset of

deforestation in LPJ-GUESS compared to the generally more open landscape in JSBACH. Thus, the differences between the

models and REVEALS in the Late Holocene trend can clearly be attributed to the incorrect appearance of anthropogenic

deforestation in the models, contributing to the overestimation of tree cover in LPJ-GUESS.

The strongest differences between the models with respect to the total tree cover occur for the British Isles, Central Europe

and southern Europe as well as the Atlantic coastal regions, in which JSBACH simulates small tree-cover fractions at all

time slices. Both models show spatial differences in the agreement with the REVEALS results. Whereas LPJ-GUESS

indicates relatively good agreement with REVEALS in Central Europe, JSBACH exhibits the strongest mismatch with the

REVEALS reconstructions for all time slices. This is partly caused by a too strong wind throw in JSBACH that substantially

reduces the simulated cover fraction of trees in large parts of Europe as shown by additional sensitivity experiments. The

strength of the effect of disturbances on the vegetation is static and calibrated to modern conditions, like most of the

parameters in vegetation models. However, the mid-Holocene natural forests were probably much more stable and less

sensitive to disturbances than the heavily human-altered present-day forests. Thus, whether these model parameter values

may be valid for the entire simulation, is questionable.

Both vegetation models fail to reproduce the tree cover changes in (mountainous) Northern Scandinavia and on the British

Isles. For the other regions, the degree of agreement varies with time. LPJ-GUESS exhibits the overall best agreement with

the REVEALS reconstructions at 6ka, while JSBACH agrees best with REVEALS at PI. In most regions, the models are not





able to simulate the correct ratio of deciduous to evergreen trees. In the mean, the distribution of evergreen trees agrees better between the models and REVEALS than the distribution of deciduous trees, except for eastern Europe. A steady improvement of the agreement of the deciduous tree cover in JSBACH with REVEALS leads to a reduction of the model-data mismatch towards PI. In contrast, the misrepresentation of land-use history in the models, i.e., a substantial anthropogenic deforestation prior to 2ka, leads to a worsening of the agreement between LPJ-GUESS and REVEALS with

time. Consequently, the model results converge towards PI.

Our study highlights the fact that model settings that are tuned for present-day conditions may be inappropriate for palaeo-simulations and complicate model-data comparisons with additional challenges. Moreover, our analysis identifies land-use as the main driver of the decrease in forest cover in Europe during the mid- and Late Holocene. Changes in climate have only a minor effect.

**Tables**

Table 1: Overview of selected time-slices and their acronyms in this study. The MPI-ESM1.2 periods used as climate forcing for the LPJ-GUESS model are provided in both, model years (1001 = 7949 BP) and corresponding calibrated [14]C years BP. The time windows for the pollen-based REVEALS-estimates of regional plant cover follow the standard protocol used in PAGES LandCover6k (Marquer et al., 2019).


| Acronym | MPI-ESM 1.2 model years | MPI-ESM period [BP] | REVEALS time window | REVEALS period [BP] |
|---------|-------------------------|---------------------|---------------------|---------------------|
| 8ka | 1090-1220 | 7860-7730 | 18 | 8200-7700 |
| 6ka | 2881-3000 | 6069-5950 | 14 | 6200-5700 |
| 4ka | 4870-5000 | 4080-3950 | 10 | 4200-3700 |
| 2ka | 6960-7078 | 1990-1872 | 6 | 2200-1700 |
| 1ka | 8000-8130 | 950-820 | 4 | 1200-700 |
| PI | 8660-8780 | 290-170 | 2 | 350-100 |





Table 2: Assignment of the plant taxa used in the pollen-based REVEALS reconstructions to the plant-functional types (PFTs) in the Dynamic Global Vegetation Models LPJ-GUESS and JSBACH.

| REVEALS taxa | LPJ-Guess PFT set | JSBACH PFT set | Land-cover type |
|---|---|---|---|
| *Abies* | Temperate needleleaved evergreen tree (TeNE) | Extra-tropical evergreen trees | |
| *Picea* | Boreal needleleaved evergreen tree | | |
| *Pinus* | Boreal shade-intolerant evergreen tree | | |
| *Carpinus* | | | |
| *Fagus* | Shade-tolerant temperate broadleaved summergreen tree | | |
| *Tilia* | | | |
| *Ulmus* | | | Forest |
| *Alnus* | | Extra-tropical deciduous trees | |
| *Betula* | | | |
| *Corylus* | Shade-intolerant broadleaved summergreen tree | | |
| *Fraxinus* | | | |
| *Quercus* | | | |
| *Salix* | | | |
| *Juniperus* | Boreal evergreen shrub | cold shrubs | Open |
| *Calluna vulgaris* | | | |
| *Artemisia* | | | |
| *Cyperaceae* | | | |
| *Filipendula* | | | |
| *Gramineae* | Cool (C3) grass (grasslands and pastures) | C3 grass (grasslands and pastures) | |
| *Plantago lanceolata* | | | |
| *Plantago media* | | | |
| *Plantago montana* | | | |
| *Rumex acetosa-t* | | | |
| *Cerealia-t* | Crops (Triticum spp., Hordeum vulgare, Secale cereale , Avena sativa, "TeWW") | Crops | |
| *Secale-t* | | | |



**Figures**

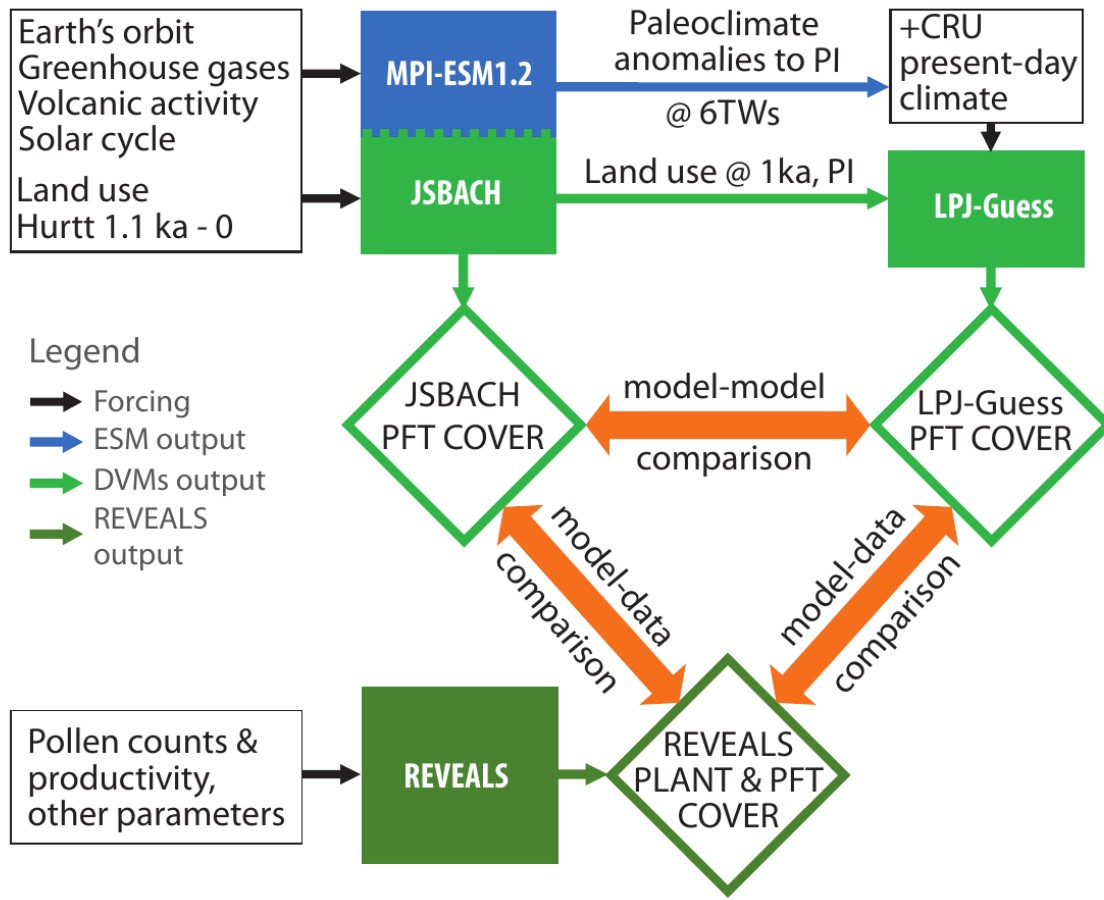

Figure 1: Flow chart of the strategy for the comparison between the plant functional type (PFT) cover simulated by the Dynamic Global Vegetation Models JSBACH (interactively coupled in the Earth System Model MPI-ESM1.2) and LPJ-GUESS (standalone model) and the pollen-based REVEALS plant-cover reconstructions. The MPI-ESM1.2 simulation and the REVEALS-based reconstructions have been published earlier in Bader et al., (2020) and Marquer et al. (2019), respectively. Within this study, LPJ-GUESS simulations for six different time-windows (TWs) were performed and compared with the MPI-ESM1.2 results and REVEALS reconstructions. As land-use forcing for JSBACH, a preliminary version of the LUH2 dataset by Hurtt et al. (2020) was used. The climate and land-use forcings for LPJ-GUESS were extracted from the output of the MPI-ESM1.2 model, but to overcome temperature biases due to the coarse spatial resolution, the MPI-ESM simulated climate anomalies to PI were interpolated bilinearly to a 0.5°x0.5° grid and added to the observational CRU-dataset (Harris et al., 2020) before prescribing them to LPJ-GUESS.

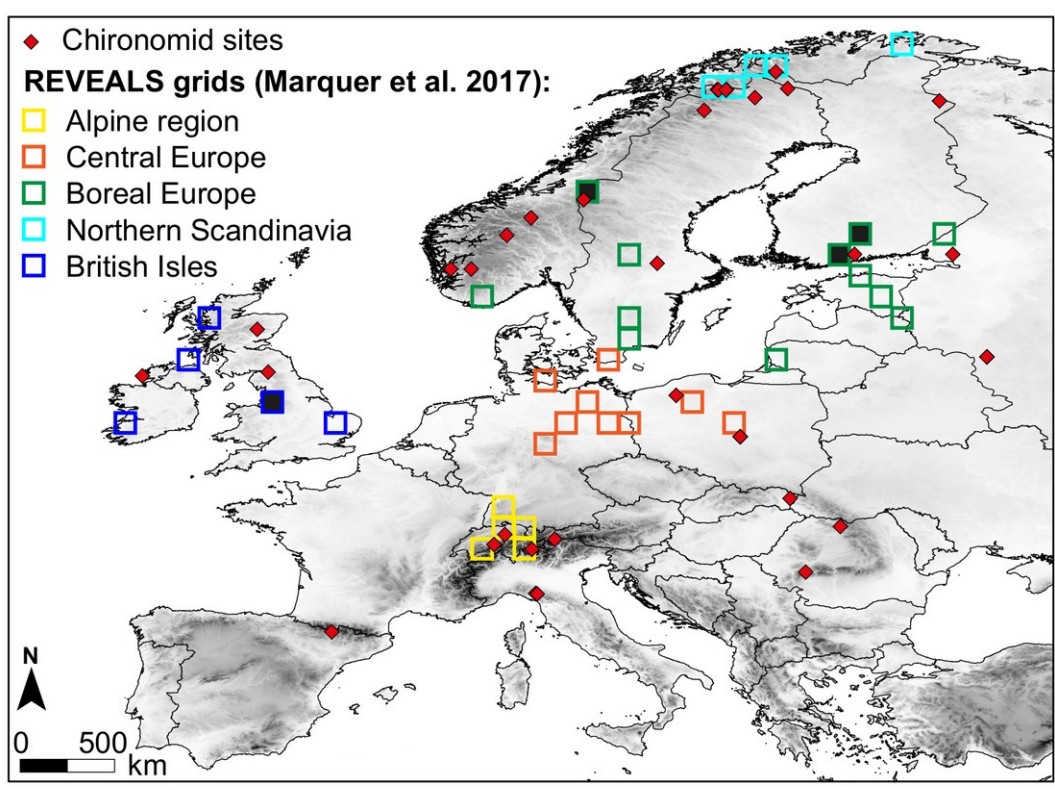

Figure 2: Grid cells with REVEALS estimates of plant abundances, grouped into five biogeographical regions (for detailed definition see Marquer et al., (2017), see figure legend for the names of the regions) and sites with chironomid data (red diamonds). Note that the four REVEALS grid cells that are colour-filled include only two small sites or one large bog (with few additional small sites) and are, thus, less reliable (cf. Sec. 2.3.2 for explanation).





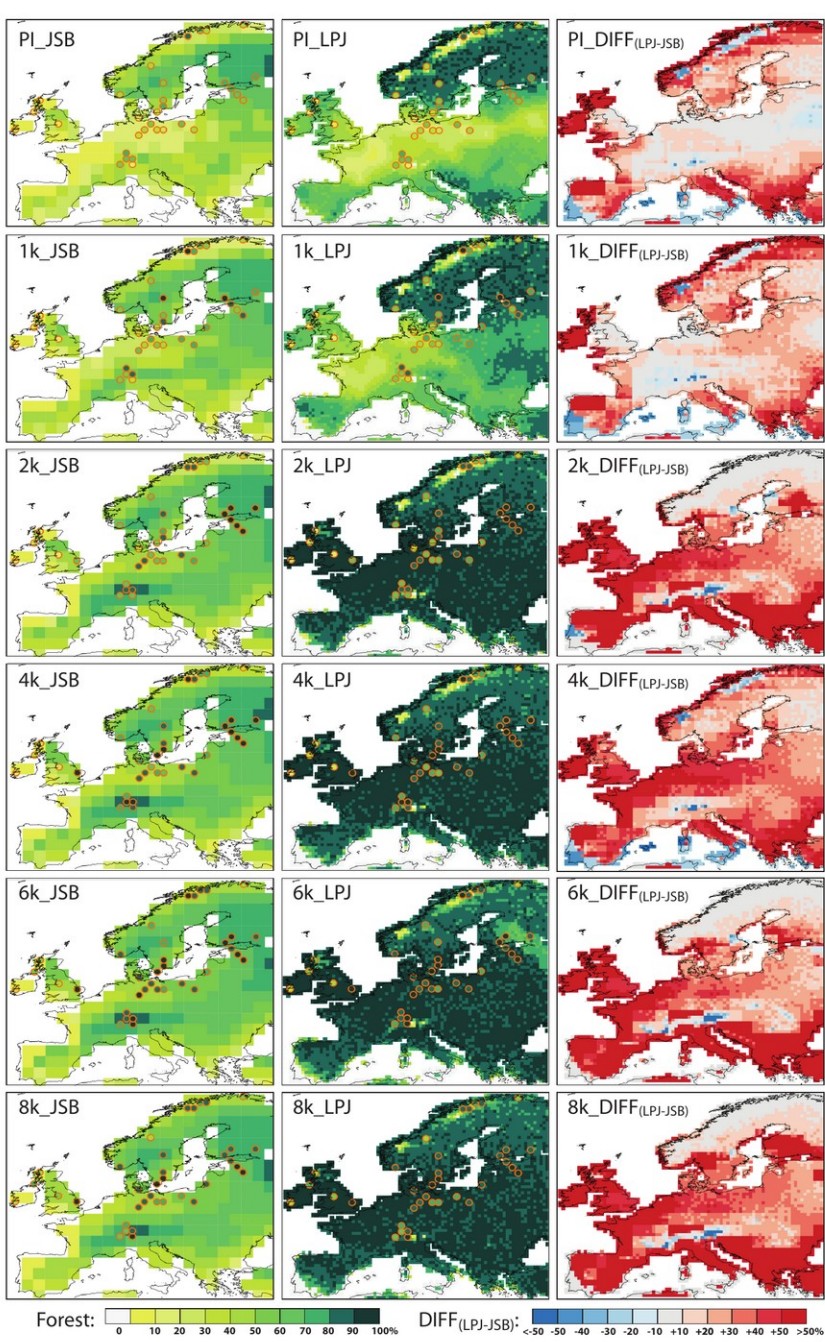

Figure 3: Total tree-cover fraction (in absolute fraction of the grid-cells) for six time-slices simulated by JSBACH (JSB) (left) and by LPJ-GUESS (LPJ) (centre) and the model difference (right). The pollen-based REVEALS tree cover is superimposed on the maps with the same colour scheme as the model results.





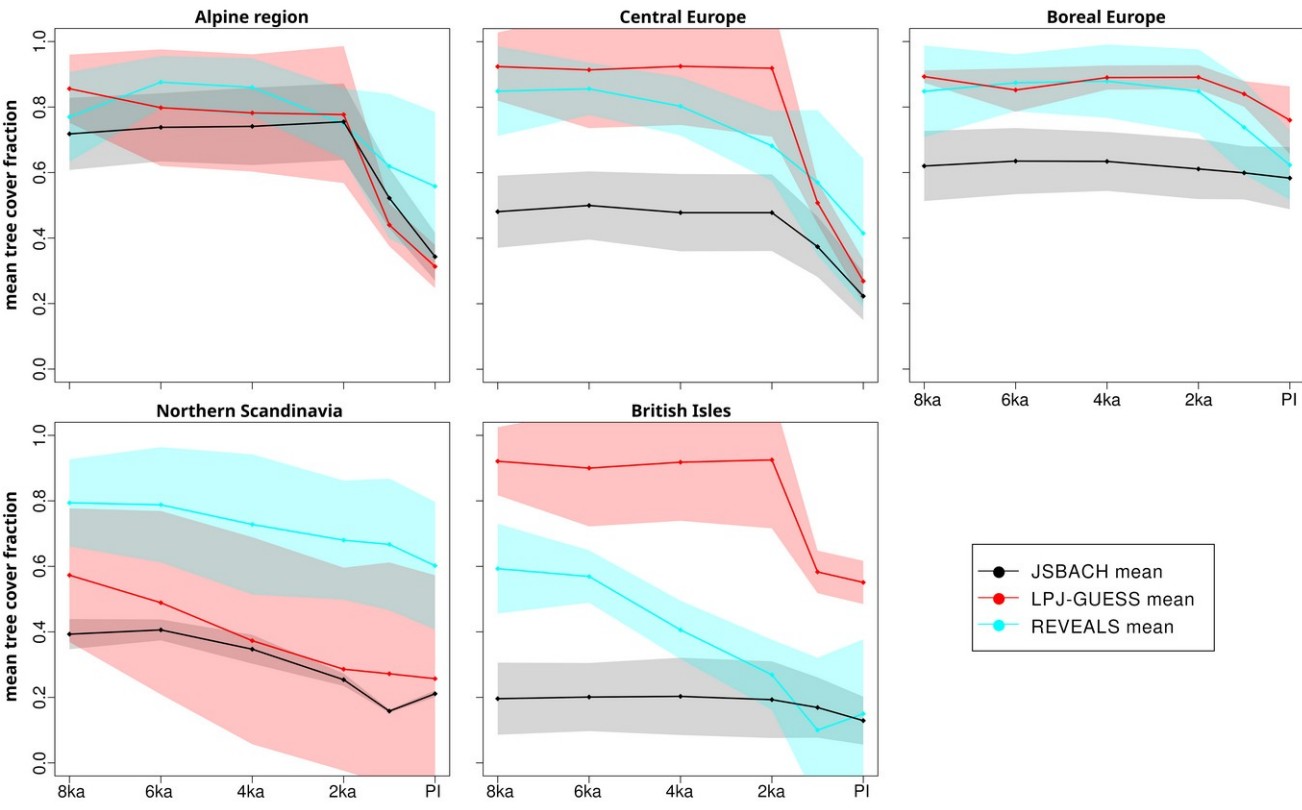

Figure 4: Model-simulated and pollen-based REVEALS mean tree-cover fraction averaged over the five regions as displayed in Fig. 2, JSBACH (black), LPJ-GUESS (red) and REVEALS (cyan). The inter grid-cell +/- standard deviation is shown with shading. Note that the standard errors on the REVEALS estimates are not considered. Time-series are plotted based on six time-slices (dots) and linearly interpolated.





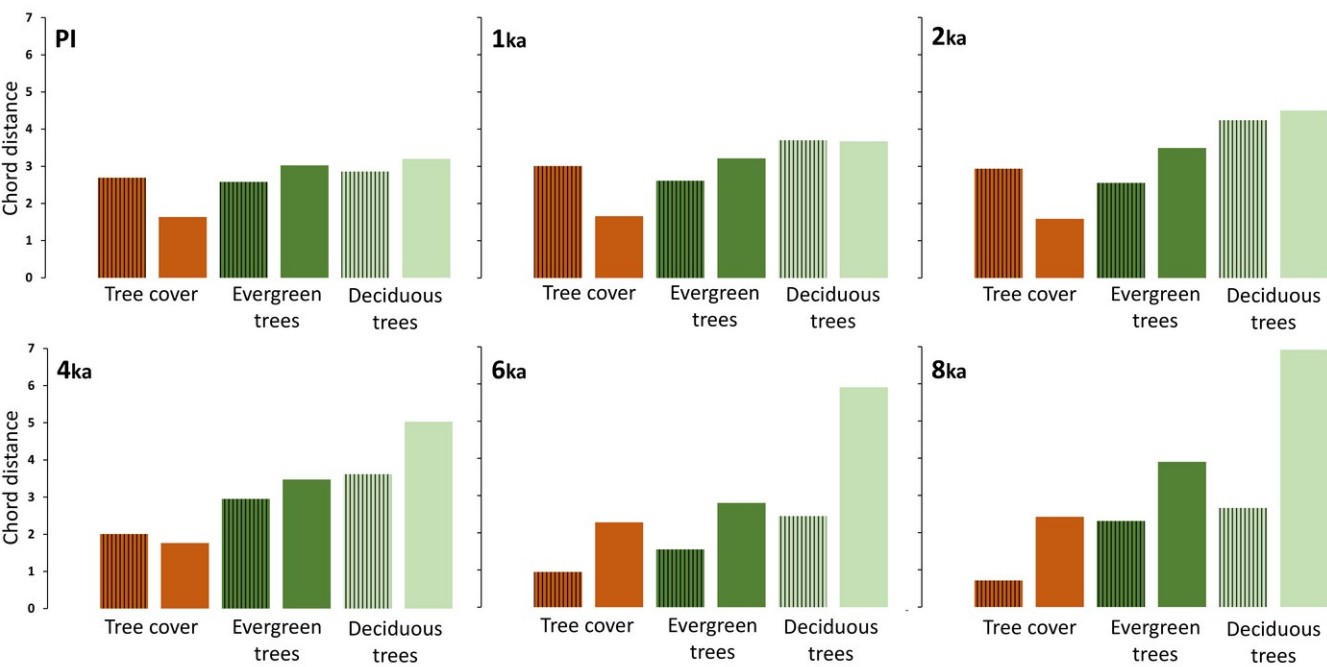

Figure 5: Chord distance between JSBACH and REVEALS (no pattern) and between LPJ-GUESS and REVEALS (vertical stripes) for total tree cover (brownish), evergreen trees (dark green) and deciduous trees (light green) for six time-slices.



Figure 6: Agreement between REVEALS and JSBACH (left) and REVEALS and LPJ-GUESS (right) for total tree cover (Forest, upper panel), evergreen trees (TET, centre) and deciduous trees (TDT, lower panel) (see legend for colours). The plot is based on a three-scale agreement index to quantify and evaluate the spatial differences between the model-simulated vegetation and the pollen-based REVEALS plant cover in each of the REVEALS grid cells (see Methods for 770 details). The six circles display the results for the six time slices, from the oldest (8ka, outermost circle) to the younger one (PI, innermost circle).



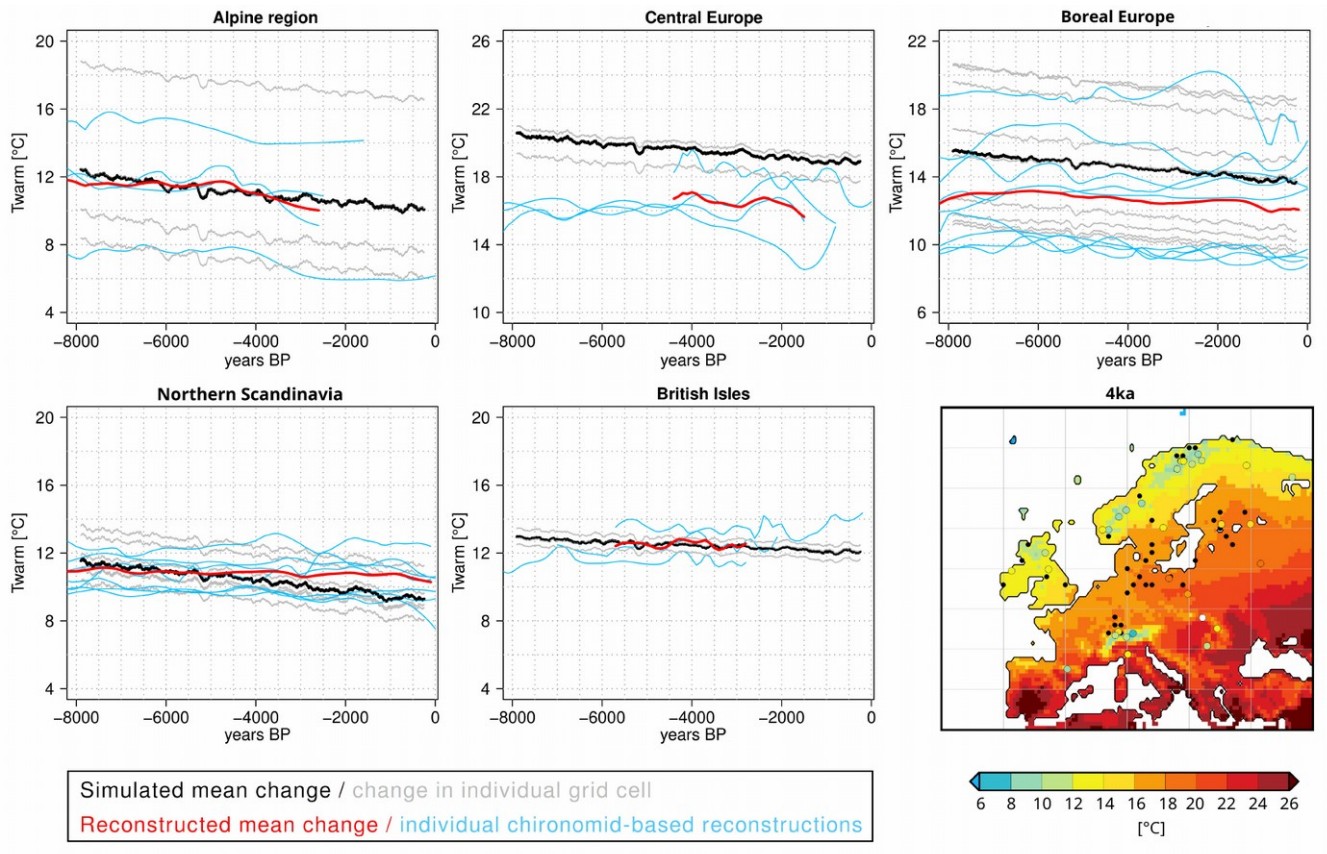

Figure 7: Comparison of the LPJ-GUESS climate forcing data (i.e. MPI-ESM1.2-simulated Holocene change in
temperature of the warmest month (Twarm, [°C]) added as anomaly (from PI) to the CRU TS 4.0 dataset (Harris et al., 2020)
and smoothed by a 200-year running mean, and chironomid-based temperature reconstructions (smoothed and interpolated
on an equally distant time-axis) for the study regions (Fig. 2) (see Tab. B1 in the Appendix B for further details on the
reconstructions). For each region, we have plotted the simulated changes in Twarm in the individual grid cells with
chironomid-inferred Twarm available (grey) and their mean (black), as well as the individual chironomid-inferred Twarm
(blue) and their mean (red). In the bottom right panel, we show the model-simulated (gridded) and chironomid-inferred
(colour dots) Twarm [°C] at 4ka. The black dots represent the location of the grid cells with pollen-based REVEALS
plant-cover reconstructions.

Figure 8: Sensitivity test with different disturbance settings. Upper nine panels: tree cover difference [in absolute % of the grid-cell area] between LPJ-GUESS simulations with an occurrence interval of 100 years (standard setup) and 25 years (upper row) or 200 years (second row) and between the JSBACH simulations in the standard setup (ST, used in this study) and with halved wind damage scaling parameter (RW) (bottom row). For each setup, the differences in cover of total tree (Forest) (left column), deciduous tree (TDT) (mid column) and evergreen tree (TET) (right column) are shown. For LPJ-GUESS, occurrence intervals of e.g. 25 years means that stochastic patch-destroying disturbance occur once per 25 simulation years. Lower panel (bar plot): difference in simulated tree-cover fractions between the simulations with standard setup and the four different disturbance-occurrence intervals (DI) in LPJ-GUESS (LPJ) and between the standard and reduced wind throw (RW) simulations in JSBACH, averaged over the entire region.



## Appendix A Evaluation of the PI climate simulated by MPI-ESM1.2

To get at least partly rid of systematic model biases such as induced by the smoothed orography in the relatively coarse model grid of MPI-ESM1.2, we have used an anomaly approach to design the climate forcing fields for LPJ-GUESS. We have added the anomaly between a certain time-slice and the pre-industrial (PI) climatological mean simulated by MPI-ESM1.2 and have added this anomaly to observations (CRU TS 4.0, period 1901-1930, Harris et al., 2020). These modified climate states have then been used as forcing for LPJ-GUESS. Thus, the vegetation models experienced the same climate dynamics during the Holocene, but have a different reference state, i.e. CRU TS4.0 observations in LPJ-GUESS but PI climate in JSBACH. To infer the differences between these basic states, we evaluate the MPI-ESM1.2 PI climate with the CRU TS 4.0 dataset (Fig. A1). The differences in temperature correlate with the orographic pattern revealing the strongest mismatch in mountainous regions. Here, MPI-ESM1.2 generally calculates higher temperatures than observed due to much lower mountain heights in the coarse resolution used in the simulation. Furthermore, simulated summer and winter temperatures are underestimated by 1-2 K in large parts of central, eastern and northern Europe compared to the observations. South of 50°N temperatures are much too high in MPI-ESM1.2 during PI, particularly during summer in the Mediterranean domain. However, in the regions for which REVEALS estimates exist in (Marquer et al., 2017), temperatures differ only slightly between the CRU observations and MPI-ESM1.2.

The annual mean precipitation is strongly overestimated by MPI-ESM1.2 in most regions of the European continent, ranging up to 950 mm/year in southern Norway and 700 mm/year in central France and Spain. Precipitation levels are way too low around the Mediterranean Sea (up to 1250 mm/year) and along the West Atlantic Coast of the British Isles (up to 2000 mm/year) and Scandinavia (up to 1000 mm/year). However, we assume that in the regions analysed in this study (i.e. the regions with REVEALS reconstructions), the vegetation dynamic is driven by the temperature signal as inferred in another study based on a slightly different MPI-ESM1.2 simulation (cf. Dallmeyer et al., 2021). In the Mediterranean area, the deficit in precipitation and too warm climate in the model probably contribute to the underestimated tree coverage, but this region is not a core part of our study here.



Figure A1: Difference between the pre-industrial climate simulated by MPI-ESM1.2 (bilinearly remapped on a 0.5 grid)
and the CRU TS 4.0 dataset (1901-1930) (Harris et al., 2020) chosen as the basic state for the LPJ-GUESS climate forcing.
Differences in summer temperature [K] (upper left), winter temperature [K] (upper right), and annual mean precipitation
[mm/year] (bottom left) are shown. Bottom right: orography (the more brownish the higher the mountains) based on the
ETOPO5 dataset (National Geophysical Data Center, 1993). The black dots display the grid-cells for which pollen-based
REVEALS estimates of plant cover are available.



# Appendix B List of the chironomid records used in this study

Table B1: List of chironomid records used in this study. Most of the chironomid-based reconstructions used in this study were extracted from the Kaufman et al. (2020) database. For these records, the data set name, site information and references (in form of the doi) were taken from the Temp12k_metadata table provided by Kaufman et al. (2020). For the other reconstructions (marked with *), the information was added accordingly.

| Region | Data Set Name | Lat. (°) | Lon. (°) | Elev.(m) | Reference |
|---|---|---|---|---|---|
| | Stazersee.Heiri.2015 | 46,50 | 9,87 | 1809 | 10.1177/0959683614556382 |
| Alpine | Hinterburgsee.Heiri.2015 | 46,72 | 8,07 | 1515 | 10.1191/0959683603hl640ft |
| Region | SchwarzseeobSoelden.Ilyashuk.2011 | 46,97 | 10,95 | 2796 | 10.1016/j.quascirev.2010.10.008 |
| | Egelsee.Larocque.2010 | 47,18 | 8,58 | 770 | 10.1007/s10933-009-9358-z |
| | Zabieniec.Plociennik.2011 | 51,85 | 19,78 | 180 | 10.1016/j.palaeo.2011.05.010 |
| | Zabieniec.Luoto.2019* | 51,85 | 19,78 | 180 | 10.3354/cr01543 |
| Central | Zabieniec.Kotrys.2020* | 51,85 | 19,78 | 180 | 10.1111/bor.12406 |
| Europe | Lake.Spore.Pleskot.2022* | 53,80 | 16,73 | <50 | 10.1016/j.palaeo.2021.110758 |
| | M25.Smolensk.Mroczkowska.2021* | 55,63 | 31,54 | <200 | 10.3390/w13111611 |
| | STIIIA.Smolensk.Płóciennik.2022* | 55,63 | 31,54 | <200 | 10.1016/j.catena.2022.106206 |
| | VestreOykjamytjorn.Velle.2005 | 59,82 | 6,00 | 594 | 10.1016/j.quascirev.2004.10.010 |
| | Holebudalen.Seppa.2009 | 59,83 | 6,98 | 1144 | 10.5194/cp-5-523-2009 |
| Boreal | Gilltjarnen.Antonsson.2006 | 60,08 | 15,83 | 172 | 10.1002/jqs.1004 |
| Europe | Hirvijaervi.Luoto.2010 | 60,51 | 25,23 | 104 | 10.1002/jqs.1417 |
| | Medvedevskoe.Nazarova.2018 | 60,53 | 29,90 | 102,2 | 10.1134/S1028334X18060144 |
| | brurskardstjorni.Velle.2005 | 61,42 | 8,67 | 1309 | 10.1016/j.quascirev.2004.10.010 |
| | Ratasjoen.Velle.2005 | 62,27 | 9,83 | 1169 | 10.1016/j.quascirev.2004.10.010 |
| | Spaime.Hammarlund.2004 | 63,12 | 12,32 | 887 | 10.1191/0959683604hl756rp |
| Northern | sjuuodjijaure.Rosen.2001 | 67,37 | 18,07 | 826 | 10.5194/cp-10-1605-2014 |
| Scandi- | AlanenLaanijarvi.Heinrichs.2005 | 67,97 | 20,48 | 365 | 10.1111/j.1502-3885.2005.tb01015.x |
| navia | VuolepNjakajaure.Heinrichs.2006 | 68,33 | 18,75 | 409 | 10.1007/s10933-006-0010-x |
| | vuoskkujavri.Bigler.2002 | 68,33 | 19,10 | 348 | 10.1016/j.quascirev.2004.04.006 |
| | 850Lake.Shemesh.2001 | 68,37 | 19,12 | 850 | 10.1191/095968301678302887 |
| | Njulla.Larocque.2004 | 68,37 | 18,70 | 999 | 10.1023/A:1022850925937 |
| | Tsuolbmajavri.Korhola.2002 | 68,41 | 22,05 | 526 | 10.1016/S0277-3791(02)00003-3 |

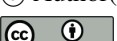



| | | | | | |
|---|---|---|---|---|---|
| | Toskaljavri.Seppa.2002 | 69,20 | 21,47 | 704 | 10.1002/jqs.678 |
| British Isles | LoughMeenachrinna.Taylor.2018 | 54,73 | -8,68 | 286 | 10.1016/j.palaeo.2018.06.006 |
| | TalkinTarn.Langdon.2004 | 54,92 | -2,71 | 130 | 10.1023/B:JOPL.0000029433.85764.a5 |
| | Lochnagar30.Dalton.2005 | 56,96 | -3,23 | 788 | 10.1016/j.palaeo.2005.02.007 |
| | BasadelaMoraLake.Tarrats.2018 | 42,54 | 0,33 | 1914 | 10.1177/0959683618788662 |
| | LagoVerdarolo.Samartin.2011 | 44,36 | 10,12 | 1390 | 10.1038/NGEO2891 |
| | Gemini.Samartin.2017 | 44,39 | 10,05 | 1349 | 10.1038/NGEO2891 |
| | TauldintreBrazi.Toth.2015 | 45,40 | 22,90 | 1740 | 10.1177/0959683614565953 |
| no specific region | TaulMuced.Diaconu.2017 | 47,57 | 24,55 | 1360 | 10.1016/j.palaeo.2017.05.007 |
| | Hypkana.Hajkova.2016 | 48,91 | 22,16 | 820 | 10.1016/j.quascirev.2016.04.001 |
| | Topptjonna.Paus.2011 | 62,38 | 9,67 | 1316 | 10.1016/j.quascirev.2011.04.010 |
| | Berkut.Ilyashuk.2005 | 66,34 | 36,66 | 25 | 10.1191/0959683605hl865ra |
| | Kharinei.Jones.2011 | 67,36 | 62,75 | 108 | 10.1007/s10933-011-9528-7 |
| | Sokli.Shala.2017 | 67,81 | 29,28 | 220 | 10.1177/0959683617708442 |

**Code and data availability**

The primary data, i.e. the model code for MPI-ESM, are freely available to the scientific community and can be accessed with a licence (https://mpimet.mpg.de/en/science/modeling-with-icon/code-availability, last access: 24 November 2021). The simulation (simulation identity: slo0021) will be published soon on the Earth System Grid.

The educational version of LPJ-GUESS is available for download (https://web.nateko.lu.se/lpj-guess/education/, last access: 06.12.2022) and a fully functional version is available for researchers from Department of Physical Geography and Ecosystem Sciences at Lund University upon request.

The REVEALS-based vegetation estimates are stored in PANGAEA (Marquer et al. (2019): https://doi.pangaea.de/10.1594/PANGAEA.900966)



In addition, secondary data and scripts that may be useful in reproducing the authors' work are archived by the Max Planck
       Institute for Meteorology and are accessible without any restrictions. The link will be published with the final version of the
       paper.

**Author contribution**

All authors planned the study and were involved in the analysis and discussion of the results. AD, AP, LM, and MJG wrote
the manuscript. AP, AD, LM and AS prepared the figures. All authors commented on, discussed, and edited the final
       manuscript.

**Competing interests**

AD is currently acting as a Guest Editor for the Climate of the Past Special Issue "Past vegetation dynamics and their role in
past climate changes".

**Acknowledgements**

This work contributes to the project PalMod, funded by the German Federal Ministry of Education and Research (BMBF),
Research for Sustainability initiative (FONA, www.fona.de). Anne Dallmeyer was financed by PalMod (Grant number:
01LP1920A). Anneli Poska was financed by Estonian Research Council grant PRG323 and by the strategic research area
BECC (Biodiversity and Ecosystem Services in a Changing Climate at Lund University; https://www.becc.lu.se/). Andrea
Seim received funding from the Swedish Research Council (Vetenskapsrådet, grant no. 2018-01272). This study is also a
contribution to the Swedish strategic research area MERGE (ModElling the Regional and Global Earth system;
www.merge.lu.se) and to the Past Global Change (PAGES) project and its working group LandCover6k
(http://pastglobalchanges.org/landcover6k) that in turn received support from the Swiss National Science Foundation, the
Swiss Academy of Sciences, the US National Science Foundation, and the Chinese Academy of Sciences. Financial support
from the Linnaeus University's Faculty of Health and Life Science is acknowledged for Marie-José Gaillard.





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
