# Peer review of "The challenge of comparing pollen-based quantitative vegetation reconstructions with outputs from vegetation models – a European perspective"

_Climate of the Past, 2023_

## Referee Comment (RC2)

The challenge of comparing pollen-based quantitative vegetation reconstructions with outputs from vegetation models – a European perspective

The authors implemented the quantitative tree-cover comparison in Europe from the middle to late Holocene through the top-down or forward (simulated estimates by the land surface model JSBASH in MPI-ESM 1.2 and the DGVM LPJ-GUESS) and bottom-up or inverse (pollen-based estimates using the REVEALS model) approaches. Concerning the temporal trends in most of Europe, tree cover was consistently greater during the middle Holocene and considerably less in the period close to the present. However, this study shows that pollen-based tree cover reduction began much earlier and was less abrupt than the top-down approach. Mismatches in tree cover between the two approaches potentially from inappropriate model settings, including parameterisation, not biases in the climate.

This study is very interesting and should be published in the Climate of the Past, but it needs refinements, additional information, and polishing to become more accessible to broader audiences. The necessary analysis has been completed, but the main text may need large rewriting. I describe the main concerns about this manuscript and then get into more detail by providing line-by-line comments, suggestions, or questions.

**General comments**
- The text is too long and could be more to the point. This manuscript has a very long discussion (Section 4), but it contains more than what should be discussed. See specific comments.
- In the introduction, previous studies on pollen-based tree cover reconstructions are completely missing. I deduce that conventional pollen analysis did not work on tree-cover reconstructions, so the authors used pollen-based tree-cover reconstruction with the REVEALS model. However, this reconstruction strongly depends on the model used (i.e., REVEALS), and the model requires several assumptions for reconstructing tree cover. The authors address the issues in the discussion section (Section 4). But, known problems and assumptions for the reconstruction should be clearly stated in the introduction (Section 1) and methodology section (Section 2), as it is the basis for this study.
- Most methods for pollen-based climate reconstructions require that vegetation is in dynamic equilibrium with climate. What is the assumed relationship between vegetation and climate in the pollen-based tree-cover reconstruction using the REVEALS model? What about a top-down approach with JSBASH and LPJ-GUESS?
- In the previous study (Hengl et al., 2018), it seems that the problem of parameterisation of DGVM, one of the conclusions of this study, was already pointed out, but was this issue not taken into account in the experimental design of this study? Or, were the two models JSBASH and LPJ-GUESS used in this study as a result of consideration?
- English grammar issues, e.g. uses of hyphens. 'land use' is noun, and 'land-use' is adjective. Please also check the use of commas. It will make the text more readable.

**Specific comments**

Abstract

L30. 'todays' to 'today' (?)

Introduction

After reading the introduction, the motivation for this study is clear, but unclear background of this study, including why it used the two models (JSBASH and LPJ-GUESS) and the pollen-based tree-cover reconstructions by the REVEALS model. Moreover, as there are probably many know

problems with the forward modelling approach and pollen-based reconstruction in previous studies, the main ones should be made explicit here.

L40. 'This requires also …' to 'This also requires …' What is this?

L47-49. 'This is one … Comparison of DGVM… ' It seems that there are two unrelated sentences. How are these sentences related? Why is it limited to the Holocene, not 'any palaeo'?

L55. 'Pollen-based vegetation reconstructions indicates …' to 'Pollen analysis indicate … (?)

L57. 'These land-use related land-cover changes …' to 'Such human-induced land-cover changes …'

L64. 'for the Early Holocene and over the last 6000 years' to 'during the Holocene'

L63-70. As mentioned above, what was the assumed relationship between vegetation and climate in the previous studies?

L75. Based on previous studies, it would be better to explain in the introduction why the authors used the two models, JSBASH and LPJ-GUESS. The authors should also briefly explain why the pollen-based tree-cover reconstructions by the REVEALS model were used instead of simple pollen analysis.

Methods

L90. '(PFT)' to '(PFTs)'

L90. 'Trees can either be tropical or temperate about bioclimatic limits and evergreen or deciduous about phenology.'

L90. 'Grassy' to 'Herbaceous' (?)

L91. '… C4 grass and the …' to '… C4 grass, and the …'

L89-92. Were 11 PFTs (8 natural PFT and 3 anthropogenic land-use types) used in this transient MPI-ESM1.2 simulation?

L123. As the other forcings, it is better to first describe what the forcing is (i.e., the anthropogenic land-use changes) and then describe the dataset, LUH2. The sentence 'This forcing begins 1100 BP, … starting at 2100 BP.' is unclear. The transient period (2100 BP to 1100 BP) is not based on LUH2? Is the anthropogenic land-use related PFTs used only after 2100 BP?

L143. The plant functional types are already defined as PFT(s) on L90. The authors do not need to repeat it.

L145. 'biomes' to 'trees' (?)

L155-160. Was there any explanation as to why it was Holocene? Is the authors' aim to evaluate the impact of anthropogenic land use or to quantitatively compare data and models on vegetation (here, tree cover)? Why do the authors need to make the equilibrium condition at each period? Please write down the assumption of this experiment first, for example, the relationship between vegetation and climate for the modelling and reconstruction. It is unclear if the way of running LPJ-GUESS is appropriate in data-model comparison.

L190. 'vegetation' to 'tree-cover' (?)

L190. This section (2.3) is unclear/confusing. First, did the authors re-perform the pollen-based tree-cover reconstructions with the REVEALS model in this study? Or did the authors use the data from any previous studies? On L225, the authors say, 'For the present study, we chose the REVEALS reconstructions from Marquer et al. (2017).' Is this about the data itself or the reconstruction method? If the authors used datasets from the previous study, the model description could be shortened. In the introduction, it would also be good to describe previous studies on tree-cover reconstructions. In reconstruing tree cover from pollen samples using the REVEALS model, does the approach require that the relationship between vegetation and climate is equilibrium?

L192-. There's something wrong with the wording. The first several sentences in this paragraph describe pollen analysis. The authors' point is that conventional pollen analysis does not adequately account for the pollen productivity of each species, so the authors used the REVEALS model. If readers get this point, the authors do not need to explain this model in detail because the REVEALS model itself was not used in this study. On the other hand, dataset details should be described.

L195-. How high-low is the temporal resolution of the pollen data?

L198-. The authors need a reference for the sentence 'Pollen productivity varies …'

L215-. Apart from the gridded pollen-based tree-cover reconstructions with REVEALS (Marquer et al. 2019), did the authors use other data described here? Why was the information written here if the authors did not use them? In the methods/data section, it is sufficient to mention only the datasets used in this study. The authors can describe previous studies on pollen-based tree-cover reconstructions with the REVEALS model in the introduction (as mentioned above).

L233. If there is more than one pollen sample in the same time window at a grid point, were they treated equally in making the gridded data? Or, were they weighted, for example, based on the lake size?

L254. The plant functional types are already defined as PFT(s) on L90. The authors do not need to repeat it.

L258. The authors are better to rewrite the sub-subject title because there may be something wrong with the wording (?)

L268. 'To evaluate …, the squared chord distance (Prentice, 1980) is calculate for each time window.' I understand what the authors try to day, but it's grammatically incorrect.

Results

Given this study's purpose, I am unsure if Figure 3 is needed in the main text. Figures 4, 5, and 6 alone would be sufficient to characterise the simulated and pollen-based tree-cover estimates over Europe since 8ka. Even if the authors discuss simulation results in areas that cannot be compared to the reconstruction, it is impossible to determine whether they are true or wrong. Descriptions and redundancies regarding regions that are not comparable to the reconstruction may be omitted from the main text as much as possible.

L280. 'natural vegetation history' to 'natural vegetation variation' (?)

L286. '… LPJ-GUESS with fewer trees' to '… LPJ-GUESS, with fewer trees'

L298. 'in most of the represented regions with highest tree cover fractions' to 'in most regions with the highest tree cover fractions'

L337. 'the inter-model spread'

L380. 'as driver' to 'as a driver'

L400. What is 'This'?

Discussion

Is it necessary to thoroughly discuss issues that had already been pointed out before this study began, for example, the 'effect of modern parametrization and tunning of the model to modern conditions' and 'a shortcoming of the REVEALS model and pollen-based plant cover reconstruction'? Shouldn't much description, especially on tree cover reconstructions (assumptions and methodology), be done in the introduction rather than in the discussion because they are known issues and should be the background to this study?

L407. 'the model agreement to' to 'the model agreement with'

L407. Remove 'that is different' (?)

L410. Remove 'the in'

L429. 'the Appendix B' to 'Appendix B'

L429. 'we compare basically the climate that is prescribed …' to 'we compare the climate prescribed …'

L430. 'We assume that summer temperature is the main climatic driver of the vegetation in the regions considered here, …' The assumption and its analysis are a vital part of the study. They should be described in the methodology section (Section 2), and the analysis results should be written in the result section (Section 3). Only real arguments should be written here (Section 4).

L437. 'in the temperate and subarctic region**s**'

L507. 'there are various technical reasons that can lead …' to 'various technical reasons can lead …'

L515. Remove 'single'

L522. '**the** establishment'

L537. 'the disregard of wetlands' to 'the disregard for wetlands'

L538. What is 'This'?

L541. About this subject, the authors introduce the following sentence in the introduction; 'the DGVM parametrization (bioclimatic limits, disturbance intervals, fire regimes, etc.) are commonly static and based on the current state of land cover although it is characterized by unstable vegetation composition due to rapidly changing natural and anthropogenic stressors (Hengl et al., 2018).' Is this a known issue? If this is a known problem, shouldn't this sensitivity experiment also be described first in Sections 2 and 3, not here?

L578. Basic information on European land cover change during the Holocene should be presented in the introduction first, as it is part of the background to this study.

L579. '**the** first traces'

L580. '… 10000 years ago an its spread' to '… 10000 years ago and its spread'

L584. 'insights of' to 'insights into'

L585. '**a** high (preferably annual) resolution'

L615. 'Many of the assumptions of the REVEALS model are violated in the "real world" and/or violated in the past, which has been described and discussed in detail earlier (references).' It is an important background for this study, and the authors should describe it in the earlier section, for example, introduction or method. The Pollen-based tree-cover reconstruction using the REVEALS model is the basis of the data-model comparison in this study.

L621. Remove 'as such'

L624. Remove 'that have been'

L626. 'Trondman et al., (2016) showed … can' to 'Trondman et al. (2016) showed …could'

L628. Remove 'do'

L637. 'advise to use' to 'advise using'

L648. What is 'This'?

L650. 'Beside' to 'Besides' (?)

L654. What is 'This'?

L661. What is 'This'?

L668. 'a few caveats of the REVEALS model and dataset used in this study can contribute …' (?)

L713. Remove 'a'

Figures

L760. It is not easy to figure out Figure 5. Maybe a table would be easier to understand than the diagram (?)

L765. It is not easy for some people to see the green-orange-red color scale in Figure 6. It may be better to change the colours.

---

## Author Comment (AC1)

**Author response to referee comments (cp-2023-16)**

**Referee #1:**

Referee: „The authors compare pollen-based reconstructions of Holocene vegetation (REVEALS) with simulated vegetation from an Earth System model (ESM) and a dynamic global vegetation model (DGVM). They find that the simulated decline in Holocene forest cover is too abrupt and late in both models and attribute this to errors in the land-use history applied in the models. The general underestimation of forest fraction in the ESM is attributed to the parameterisation of disturbance.

This is interesting study with an important finding which is very relevant for this journal and which is strengthened by the use of two very different modelling approaches. This and the level of detail is a strength. My main comment (see below) is that the paper is overly long and the authors should aim to substantially shorten the text in several places in order to make it more accessible."

Authors: We thank Referee#1 for his/her constructive and encouraging comments that have helped to strengthen the manuscript. We substantially shortened the text according to his/her suggestions.

Main comments:

Referee: I appreciate that this is a technical paper probably mostly aimed at those within the field, but I found the text in places hard to digest. In the methods I would recommend that you include some summary paragraphs at the start of the section 2 to very briefly explain the differences between MPI-ESM, JSBACH, LPJ-GUESS and REVEALS. More generally I believe that you can make the text more concise, especially section 2 (currently ~ 6 pages), section 3.1 (2 1/2 pages), 4.1 (3 pages) and section 4.5 (2 pages). Some of the detail in section 2 could also optionally be moved to an appendix.

Authors: Thank you for this comment. We insert the following short summary of the used models to section 2: „*In this study, we use DGVMs with very different modelling approaches. JSBACH is the land-surface component of the comprehensive Earth System model MPI-ESM, and thus calculates interactively the grid-cell cover fractions of different plant functional types (PFTs) in line with the simulated climate (Reick et al., 2021). In contrast, LPJ-GUESS is a so-called "gap" model that calculates the growth of individuals or cohorts in patches, according to a prescribed climatic forcing (Smith et al., 2001). While we explore one of the few worldwide existing transient Holocene Earth System model simulations (MPI-ESM), the high spatial resolution time-slice experiments in LPJ-GUESS have been conducted in this study. LPJ-GUESS is used here because it is well-tested for the European domain (e.g. Hickler et al., 2012). The results of both models are compared to the pollen-based REVEALS reconstructions of plant cover for Europe from Marquer et al. (2017). These reconstructions were chosen before others based on other methods such as MAT (e.g. Davis et al., 2015) or pseudobiomization (e.g. Fyfe et al., 2010) because the REVEALS model has been shown to be the best approach to produce reliable quantitative reconstructions of forest cover and the only method available to date to reconstruct the cover of individual plant taxa (e.g. Hellman et al., 2008; Roberts et al., 2018; see also Introduction above). The methodological strategy used in this study is summarised in the flow chart of Figure 1.*"*

We shortened the method section and section 4.5 and moved one paragraph of section 4.5 to the method part of the REVEALS model, according to the suggestions of Referee#1 and #2. However, since climate biases in the models are commonly used as explanation for model-data mismatches, the climate comparison and the resulting exclusion of this argument for the main model-data differences in this study, are an essential part of our manuscript. Therefore, we kept the detailed discussion of the climate (sec.4.1) and description of our results (sec. 3.1).

Referee: Can the authors comment on whether other parameterisations besides the ones explored (disturbance and windthrow) could contribute the biases in the forest cover simulations?

Authors: All parameterisations can have a greater or lesser effect on the simulated climate and vegetation, but it is difficult to say how strong the effects are without running sensitivity simulations for individual parameters. In this study and for determining the causes of differences in forest cover estimates, the prescribed disturbances were the most obvious place to start because, besides natural mortality, the disturbances have a direct impact on simulated forest cover by reducing (or removing) the tree PFTs. In addition, some areas of Europe are often exposed to strong winds. In these regions, the wind-throw is particularly effective.

Minor Comments

Referee: Line 19: specify that LPJ-GUESS is the dynamic vegetation model.

Authors: We further specify the type of models.

Referee: Line 297: I could not see a figure with these results, please can you check this?

Authors: The REVEALS reconstructions are displayed as dots on top of the modelling results. We now specify in the text that these results are shown in Fig. 3.

Referee: Figure 5: could you add a key to this figure to make it easier to read?

Authors: We added labels to the bars on the figure.

Referee: Lines 588-600: It is probably worth clarifying here that the KK10 reconstruction has substantially earlier and more widespread land-use than LUH2.

Authors: Thank you for this remark. Nevertheless, we decide to keep the text as it is, because we already mention in the previous sentence (starting L594) that the land-use products differ substantially: „*However, KK10, HYDE and other ALCCs exhibit large discrepancies in their estimates of the starting time, spatial pattern and intensity of anthropogenic land-cover change, making it a challenge to simulate the effect of human-induced vegetation changes with DGVMs* (Kaplan et al., 2017; Gaillard et al., 2010).*"*

Referee: Line 615: I cannot comment on how topography is treated in REVEALS. As written this sounds a bit uncertain. Can you confirm that topography is not accounted for and how this would influence the model results. At the moment it sounds as if you're unsure whether it does or not.

Authors: The REVEALS model does not take into account topography. See discussion of this issue in Marquer et al. (2020). The authors conclude from their pilot study in the Pyrenees that the REVEALS model does not work well in mountains unless the number and location of sites capture pollen assemblages from various altitudes, below, at and above the tree line. The latter is mostly not realized in the grid-cell based REVEALS reconstructions we are using, therefore the REVEALS estimates in e.g. the Scandinavian mountains should be considered as less reliable.

We now write: *"Because topography is not accounted for in the REVEALS model and reliable reconstructions in mountainous regions require a large number of pollen records from large lakes representing the major altitudinal vegetation zones (Marquer et al., 2020), the REVEALS estimates used here need to be considered as uncertain in the Scandinavian mountains and the Alps, which may explain discrepancies with DGVMs at the grid-cell scale level in these areas (Fig.3)"*

Referee: Line 705: This seems to contradict the sensitivity test with increased disturbance you present earlier, can you comment?

Authors: We are not sure if we understand the referee's concern. In the sensitivity study, we basically analyse the effect of reduced wind throw. To further clarify our statement, we modified the text as follows: *„However, the mid-Holocene natural forests were probably much more stable and less sensitive to disturbances than the heavily human-altered present-day forests. This would justify and require the use of a lower storm vulnerability of forest in JSBACH for this period. Thus, whether the modern climate-derived model parameter values may be valid for the entire transient simulation, is questionable."*

Line 717: This seems to echo results by Kaplan et al 2017 (*Land*). Is that correct? If so, can comment on this here and earlier in the text?

Authors: Kaplan et al. (2017) evaluated the performance of two commonly used anthropogenic land-cover change models (ALCC scenarios) by comparing their estimates with REVEALS reconstructions of open land cover, and discussed the discrepancies in the light of human-habitation-related unknowns in model parametrization and spatial allocation of deforestation. Here, we discuss a different problem, the observed discrepancies in forest fraction estimates of two vegetation models and REVEALS in Europe. What is said in our conclusion has been suggested by previous papers. But no earlier analyses have looked into that specifically.
To further clarify this, we modified our conclusions as follows:
*"Our study highlights the fact that model settings that are tuned for present-day conditions may be inappropriate for palaeo-simulations and complicate model-data comparisons with additional challenges. Moreover, our analysis identifies land use as the main driver of the decrease in forest cover in Europe during the mid- and Late Holocene, as has been suggested by pollen studies and the more recent efforts to quantify pollen-inferred changes in plant cover (e.g. Robert et al., 2018) and the various scenarios of anthropogenic land-cover change developed over the last ca. 20 years (e.g. HYDE and KK10, see synthesis in Gaillard et al., 2010). Mid- and Late Holocene changes in climate have only a minor effect on forest cover, although changes in cover and distribution of individual plant taxa depend on both land use and climate (Marquer et al., 2017)."*

---

## Author Comment (AC2)

**Author response to referee comments (cp-2023-16)**

**Referee#2**

Referee: The authors implemented the quantitative tree-cover comparison in Europe from the middle to late Holocene through the top-down or forward (simulated estimates by the land surface model JSBASH in MPI-ESM 1.2 and the DGVM LPJ-GUESS) and bottom-up or inverse (pollen-based estimates using the REVEALS model) approaches. Concerning the temporal trends in most of Europe, tree cover was consistently greater during the middle Holocene and considerably less in the period close to the present. However, this study shows that pollen-based tree cover reduction began much earlier and was less abrupt than the top-down approach. Mismatches in tree cover between the two approaches potentially from inappropriate model settings, including parameterisation, not biases in the climate.

This study is very interesting and should be published in the Climate of the Past, but it needs refinements, additional information, and polishing to become more accessible to broader audiences. The necessary analysis has been completed, but the main text may need large rewriting. I describe the main concerns about this manuscript and then get into more detail by providing line-by-line comments, suggestions, or questions.

Authors: We thank Referee#2 for carefully reading the manuscript and helpful comments that improve the manuscript.

**General comments**

Referee: The text is too long and could be more to the point. This manuscript has a very long discussion (Section 4), but it contains more than what should be discussed. See specific comments.

Authors: The discussion of the potential reasons behind the model-data mismatches is the central part of our study. In our opinion, this in-depth discussion is a strength of our study because - at least to our knowledge - it is the first time that these issues are systematically evaluated in a paleoclimatic and paleoenvironmental context. We therefore decided to keep the detailed discussion section. However, we substantially shortened the method section and the discussion of the REVEALS caveats, according to the specific referee comments below.

Referee: In the introduction, previous studies on pollen-based tree cover reconstructions are completely missing. I deduce that conventional pollen analysis did not work on tree-cover reconstructions, so the authors used pollen-based tree-cover reconstruction with the REVEALS model. However, this reconstruction strongly depends on the model used (i.e. REVEALS), and the model requires several assumptions for reconstructing tree cover. The authors address the issues in the discussion section (Section 4). But, known problems and assumptions for the reconstruction should be clearly stated in the introduction (Section 1) and methodology section (Section 2), as it is the basis for this study.

Authors: We added references on previous pollen-based tree cover reconstructions in the introduction. We used the REVEALS based reconstructions, because it is the only method available to date that can account for inter-taxonomic differences in pollen productivity and dispersal and estimate plant abundance in percentage cover. We now explain the motivation for using REVEALS in the introduction and in a summary paragraph right at the beginning of the method section (see response to specific comment on this below). As suggested, we moved parts of the discussion section to the REVEALS method section.

We wrote in the Introduction: *"Among existing empirical proxies of past vegetation, pollen records from lake sediments or peat deposits have the best potential for quantitative reconstructions of plant abundance or spatial cover. However, pollen records are subject to several shortcomings such as unknown size of source area of pollen and differences in pollen productivity and dispersal properties between plant taxa (e.g. Prentice, 1985). These issues imply that pollen percentages (and pollen accumulation rates) from fossil pollen assemblages can only provide qualitative or semi-quantitative information on past vegetation changes, i.e. sporadic or regular presence, occurrence in more or less large quantities, and increases and decreases in abundance of plant taxa. Different methods have been developed to overcome these problems and reconstruct plant cover, e.g. biomization, the modern analogue technique (MAT), pseudo-biomization, and the landscape reconstruction algorithm (REVEALS and LOVE models). These methods are described and evaluated in e.g. Hellman et al. (2008) and Roberts et al. (2018).*

*The model REVEALS (Regional Estimates of Vegetation Abundance from Large Sites (Sugita, 2007) is the only method so far that accounts for inter-taxonomic differences in pollen productivity and dispersal and deposition properties and provides estimates of plant cover (in % cover of a defined area) for individual taxa. In recent years, datasets of pollen-based REVEALS plant cover were produced at a 1° grid-cell spatial scale for large regions of the world, i.e. Europe, China and N America-Canada (Trondman et al., 2015; Marquer et al., 2017; Dawson et al., 2018; Cao et al., 2019; Githumbi et al. 2022a, Li et al, 2023; Serge et al., 2023). These datasets are appropriate for use in paleoclimate modelling (e.g. Strandberg et al., 2021, 2023) and evaluation of dynamic vegetation models (e.g. Marquer et al., 2017; 2018) and scenarios of anthropogenic land-cover change (e.g. Kaplan et al., 2017)."*

Referee: Most methods for pollen-based climate reconstructions require that vegetation is in dynamic equilibrium with climate. What is the assumed relationship between vegetation and climate in the pollen-based tree-cover reconstruction using the REVEALS model? What about a top-down approach with JSBASH and LPJ-GUESS?

Authors: REVEALS does not require assumptions on the relationship between vegetation and climate. The REVEALS model does not use climate parameters at all. It uses parameters related to the morphology of pollen and the pollen productivity of plants, as well as models of the transport of small particles in the air. However, it uses pollen transportation related parameters such as average wind speed and some atmospheric conditions. It is a "translation" of pollen % into plant % cover based on values of pollen productivity for the plants we find as pollen in palaeo archives and models of the transport of pollen from the mother plants to the deposition site (lake or bog) . JSBACH is part of the fully coupled atmosphere-ocean-land-surface model MPI-ESM. The vegetation cover is calculated dynamically in the model, in response to climatic (and ecological) changes. LPJ-GUESS has been forced by climate fields, simulated by MPI-ESM. The DGVMs, thus, simulated vegetation in quasi-equilibrium to the climate, whereas the pollen-based reconstructions may be affected by lags in the response of the vegetation to the climate signal (see e.g. Dallmeyer et al., 2022, https://doi.org/10.1038/s41467-022-33646-6).

We further clarified this in the beginning of the method part: *"In this study, we use DGVMs with very different modelling approaches. JSBACH is the land-surface component of the comprehensive Earth System model MPI-ESM, and thus calculates interactively the grid-cell cover fractions of different plant functional types (PFTs) in line with the simulated climate (Reick et al., 2021). In contrast, LPJ-GUESS is a so-called "gap" model that calculates the growth of individuals or cohorts in patches, according to a prescribed climatic forcing (Smith et al., 2001)."*

Referee: In the previous study (Hengl et al., 2018), it seems that the problem of parameterisation of DGVM, one of the conclusions of this study, was already pointed out, but was this issue not taken into account in the experimental design of this study? Or, were the two models JSBASH and LPJ-GUESS used in this study as a result of consideration?

Authors: All models share the problem of static parameterizations and this problem cannot be solved with respect to paleo simulations, because most parameters cannot be reconstructed and certainly not for the entire globe which would be necessary to adapt the (global) Earth System models. Furthermore, running paleo simulations is expensive and requires strong computational power. The transient simulation used in this study ran for nearly one year on the super-computer of the DKRZ. These kinds of simulations cannot be repeated to test the effects of different parameter settings. In this study, we therefore "only" use one of the few (worldwide) existing transient simulations. We have not performed it for this study.

Referee: English grammar issues, e.g. uses of hyphens. 'land use' is noun, and 'land-use' is adjective. Please also check the use of commas. It will make the text more readable.

Authors: Thank you for this advice. In the manuscript, we followed the rule of adding a hyphen if more than two nouns are standing in a row. If there are less than 2 nouns in a row, no hyphen is inserted. These rules are the advice of two well-known palaeoecologists, one native British and one native American. We will discuss this issue with the editorial office. We carefully checked the manuscript for grammar and spelling mistakes again. We trust the Journal for making a final language check and decide what spelling and grammatical rules are preferred.

**Specific comments**

Abstract

Referee: L30. 'todays' to 'today' (?)
Authors: We changed it to "today's".

Introduction

Referee: After reading the introduction, the motivation for this study is clear, but unclear background of this study, including why it used the two models (JSBASH and LPJ-GUESS) and the pollen-based tree-cover reconstructions by the REVEALS model. Moreover, as there are probably many known problems with the forward modelling approach and pollen-based reconstruction in previous studies, the main ones should be made explicit here.

Authors: Thank you, we see this point. The main arguments for using JSBACH and LPJ-GUESS are a) the two models are both well-tested, also for paleoclimatic conditions and b) we could use an existing transient Holocene simulation performed in MPI-ESM that includes JSBACH. We added the motivation for using these vegetation models and REVEALS at the beginning of the method section following the suggestion by Referee#1. We now write:
*„In this study, we use DGVMs with very different modelling approaches. JSBACH is the land-surface component of the comprehensive Earth System model MPI-ESM, and thus calculates interactively the grid-cell cover fractions of different plant functional types (PFTs) in line with the simulated climate (Reick et al., 2021). In contrast, LPJ-GUESS is a so-called "gap" model that*

*calculates the growth of individuals or cohorts in patches, according to a prescribed climatic forcing (Smith et al., 2001). While we explore one of the few worldwide existing transient Holocene Earth System model simulations (MPI-ESM), the high spatial resolution time-slice experiments in LPJ-GUESS have been conducted in this study. LPJ-GUESS is used here because it is well-tested for the European domain (e.g. Hickler et al., 2012). The results of both models are compared to the pollen-based REVEALS reconstructions of plant cover for Europe from Marquer et al. (2017). These reconstructions were chosen before others based on other methods such as MAT (e.g. Davis et al., 2015) or pseudobiomization (e.g. Fyfe et al., 2010) because the REVEALS model has been shown to be the best approach to produce reliable quantitative reconstructions of forest cover and the only method available to date to reconstruct the cover of individual plant taxa (e.g. Hellman et al., 2008; Roberts et al., 2018; see also Introduction above). The methodological strategy used in this study is summarised in the flow chart of Figure 1.“*

Furthermore, we moved the part of the caveats of pollen records (in a slightly shorter version) to the introduction to better explain why we use REVEALS (see comment above).

Referee: L40. 'This requires also ...' to 'This also requires ...' What is this?
Authors: done. „This“ refer to „our ability to correctly understand the interactions between vegetation and climate“.

Referee: L47-49. 'This is one ... Comparison of DGVM... ' It seems that there are two unrelated sentences. How are these sentences related? Why is it limited to the Holocene, not 'any palaeo'?
Author: Thank you, this was indeed misleading. We have made the second sentence into a new paragraph and deleted „Holocene“.

Referee: L55. 'Pollen-based vegetation reconstructions indicates ...' to 'Pollen analysis indicate ... (?)
Authors: correction done.

Referee: L57. 'These land-use related land-cover changes ...' to 'Such human-induced land-cover changes …'
Authors: done.

Referee: L64. 'for the Early Holocene and over the last 6000 years' to 'during the Holocene'
Authors: We have changed the sentence to: *„An earlier evaluation of the performance of the DGVM LPJ-GUESS by comparing model-simulated land cover with pollen-based plant-cover reconstructions in Europe has shown clear differences between the two for the first one to two millennia of the Holocene and over the last 7000-6000 years (e.g. Marquer et al., 2017; 2018).“*

Referee: L63-70. As mentioned above, what was the assumed relationship between vegetation and climate in the previous studies?
Author: Neither the REVEALS model nor the DGVMs used in this study (and also in previous studies) need any specific assumptions on the climate, beside the bioclimatic limitations that are part of (nearly all) DGVM and represent the bioclimatic tolerance of PFTs. These bioclimatic limitations are a central part of the model physics and described in the documentations of the models. We have not changed them in this study.

Referee: L75. Based on previous studies, it would be better to explain in the introduction why the authors used the two models, JSBASH and LPJ-GUESS. The authors should also briefly explain why the pollen-based tree-cover reconstructions by the REVEALS model were used instead of simple pollen analysis.

Authors: We now explain the motivation in a summary paragraph right at the beginning of the method section (see comment above). The major motivation to use pollen-based REVEALS estimates of plant cover is of course the advantage to compare DGVM simulated plant cover (in % cover of a defined area) with REVEALS estimated plant cover (in % cover of a defined area). Comparing plant % cover with pollen % does not make much sense if we aim at making a quantitative comparison. Pollen % does not correspond to the related plant % cover due to inter-taxonomical differences in pollen productivity, dispersal and deposition properties.

**Methods**

Referee: L90. '(PFT)' to '(PFTs)'
Authors: done.

Referee: L90. 'Trees can either be tropical or temperate about bioclimatic limits and evergreen or deciduous about phenology.'
Authors: It is not only the bioclimatic limitations that distinguish tropical from extratropical tree PFTs. The PFTs are also in complex competition with each other. To avoid misunderstandings, we rephrased the PFT description:
"In this module, natural vegetation is represented by eight plant functional types (PFTs). Trees (four PFTs) can either be tropical or extratropical (i.e. boreal + temperate), and evergreen or deciduous.

The open land cover is represented by two herbaceous PFTs (C3 and C4 grass) and two shrub PFTs (raingreen shrubs and cold resistant shrubs). Land use (anthropogenic vegetation) is included as three PFTs (i.e. C3 pasture, C4 pasture, and crops). Therefore, total plant cover is represented by eleven PFTs (see details on the implementation of anthropogenic PFTs below)."

Referee: L90. 'Grassy' to 'Herbaceous' (?)
Authors: done. See author answer above.

Referee: L91. '... C4 grass and the ...' to '... C4 grass, and the …'
Authors: This sentence has changed, see author answer above.

Referee: L89-92. Were 11 PFTs (8 natural PFT and 3 anthropogenic land-use types) used in this transient MPI-ESM1.2 simulation?
Authors: Yes, in total the cover fractions of 11 PFTs have been calculated. This is now clarified in the text, see comment above (comment to L90)

Referee: L123. As the other forcings, it is better to first describe what the forcing is (i.e. the anthropogenic land-use changes) and then describe the dataset, LUH2. The sentence 'This forcing begins 1100 BP, ... starting at 2100 BP.' is unclear. The transient period (2100 BP to 1100 BP) is not based on LUH2? Is the anthropogenic land-use related PFTs used only after 2100 BP?
Authors: Thank you for the detailed questions. The LUH2 dataset starts at 850 CE, but at 850 CE, the land cover is already a mixture of anthropogenic and natural PFTs. In order to avoid an abrupt change from "no land use" to "occurrence of land use", we implemented a transition period to slowly build up the cover fractions of the anthropogenic land cover types in the model from ca. zero cover at 2100 BP to LUH2 cover at 850 CE. Since the number of PFTs cannot be changed during a

simulation, the 11 PFTs (8 natural + 3 land-use types) have been used for the entire simulation but the cover fractions of the land-use types are ca. zero before 2100 BP.

To further clarify this forcing, we rephrased the text as indicated below. However the text was moved to the previous section (we merged the two MPI-ESM method parts to one), following other referee advices.

We now write: *"Human-induced land-cover changes affect the simulation only for the last ~2000 years. In the simulation, the anthropogenic land-cover changes have been prescribed from a preliminary version of the LUH2 dataset (Hurtt et al., 2020). This dataset provides the land-use changes for the 850CE to 2100CE period and could therefore only be used as forcing of the model for the last 1000 years of the transient simulation (from 1.1 ka on). To slowly build up the land use in the course of the simulation, from zero to the anthropogenic influenced land cover at 1.1 ka, a transition period of 1000 years has been implemented starting at 2.1 ka. Land use is read…."*

Referee: L143. The plant functional types are already defined as PFT(s) on L90. The authors do not need to repeat it.
Authors: Thank you. We changed it.

Referee: L145. 'biomes' to 'trees' (?)
Authors: „biomes" is correct.

Referee: L155-160. Was there any explanation as to why it was Holocene? Is the authors' aim to evaluate the impact of anthropogenic land use or to quantitatively compare data and models on vegetation (here, tree cover)? Why do the authors need to make the equilibrium condition at each period? Please write down the assumption of this experiment first, for example, the relationship between vegetation and climate for the modelling and reconstruction. It is unclear if the way of running LPJ-GUESS is appropriate in data-model comparison.

Author: The experiment was conducted to compare reconstructed and model simulated estimates of land-cover composition before the start of human land use (8ka) and during the period of increasing intensity in anthropogenic deforestation (6ka - modern) in Europe, and to discuss the probable causes of the determined discrepancies. As models are built to mimic potential natural vegetation we assume that smallest discrepancies occur prior to anthropogenic deforestation and are reduced by including the land-use forcing. We analyse the Holocene period, because we use an existing transient MPI-ESM simulation for this period and REVEALS reconstructions are available for this period.

LPJ-GUESS is an offline vegetation model that has to be forced by a prescribed climate. In this study, we used the climate simulated in the transient (fully coupled) MPI-ESM simulation. Thus, the climate forcing used for LPJ- GUESS is not an equilibrium climate state, but rather a short (ca. 100 years) dynamic climate period. However, to ensure model-data comparability and in order to reduce the likelihood that the simulation represents extreme events rather than a typical climate for the period, we chose relatively stable climate periods close to target time (e.g. 8ka) as forcing for LPJ-GUESS runs, without major climatic fluctuations (caused by for example volcanic eruptions etc.). Like all DGVMs, LPJ-GUESS requires a spin-up period (usually run for 300 or more years using randomised combination of first few decades of climate forcing dataset) to reach quasi-equilibrium between climate, vegetation and biogeochemistry, to avoid a strong drifting of the simulated vegetation.

The REVEALS model does not require any assumptions on the vegetation-climate relationships (see detailed answer above). In LPJ-GUESS and JSBACH, bioclimatic limitation rules are implemented for each PFT that represent their climatic tolerance. The response of the vegetation in

LPJ-GUESS to the prescribed climate is part of the dynamic equations of the model, considering these bioclimatic limitations. A detailed description of the model equations goes beyond the scope of this paper, but we refer to the documentations of the model in the manuscript.

Referee: L190. 'vegetation' to 'tree-cover' (?)
Authors: The REVEALS reconstruction does not only provide tree-cover fractions, but also herbaceous plant-cover fractions. We changed the heading to: „*Pollen-based REVEALS plant-cover reconstructions*"

Referee: L190. This section (2.3) is unclear/confusing. First, did the authors re-perform the pollen-based tree-cover reconstructions with the REVEALS model in this study? Or did the authors use the data from any previous studies? On L225, the authors say, 'For the present study, we chose the REVEALS reconstructions from Marquer et al. (2017).' Is this about the data itself or the reconstruction method? If the authors used datasets from the previous study, the model description could be shortened. In the introduction, it would also be good to describe previous studies on tree-cover reconstructions. In reconstructing tree cover from pollen samples using the REVEALS model, does the approach require that the relationship between vegetation and climate is equilibrium?
Authors: We indeed just use the dataset of REVEALS-based reconstructions by Marquer et al. (2017). We rephrased this part and substantially shortened it. We added references to previous studies on tree-cover reconstructions to the introduction. REVEALS does not require any assumptions on climate-vegetation equilibrium (see detailed answers above).

Referee: L192-. There's something wrong with the wording. The first several sentences in this paragraph describe pollen analysis. The authors' point is that conventional pollen analysis does not adequately account for the pollen productivity of each species, so the authors used the REVEALS model. If readers get this point, the authors do not need to explain this model in detail because the REVEALS model itself was not used in this study. On the other hand, dataset details should be described.
Authors: We agree, we deleted the text on the pollen analysis here and partly moved it to the introduction. We furthermore substantially shortened the method part on the REVEALS model, focussing on the assumptions that are relevant for this study.

Referee: L195-. How high-low is the temporal resolution of the pollen data?
Authors: The temporal resolution of the pollen data varies a lot through time and between pollen records, from < 100 years to maximum ca. 1000 years. The temporal resolution of the REVEALS estimates is constant, i.e. 500 years from 700 to 11700 BP and 0-100, 100-350, 350-700 BP for the recent times….. Please notice, that the sentence in L195 was deleted.

Referee: L198-. The authors need a reference for the sentence 'Pollen productivity varies …'
Authors: We deleted the part on pollen here and moved it (in a shorter version) to the introduction. We now write:"*However, pollen records are subject to several shortcomings such as unknown size of source area of pollen and differences in pollen productivity and dispersal properties between plant taxa (e.g. Prentice, 1985).*"

Reference: "Prentice, I.C.: Pollen representation, source area, and basin size: toward a unified theory of pollen analysis. Quat. Res. 23 (1), 76–86, 1985."

Referee: L215-. Apart from the gridded pollen-based tree-cover reconstructions with REVEALS (Marquer et al. 2019), did the authors use other data described here? Why was the information written here if the authors did not use them? In the methods/data section, it is sufficient to mention

only the datasets used in this study. The authors can describe previous studies on pollen-based tree-cover reconstructions with the REVEALS model in the introduction (as mentioned above).
Authors: We agree and shortened this part. These datasets are already mentioned in the introduction.

Referee: L233. If there is more than one pollen sample in the same time window at a grid point, were they treated equally in making the gridded data? Or, were they weighted, for example, based on the lake size?
Authors: The gridded reconstructions are based on one to several pollen samples per study site, and on pollen sample(s) from one to several study sites. We do not weight the pollen data according to lake size, but lake size is used in the REVEALS model as it is indeed a parameter that influences pollen loading in a lake or on a bog (Sugita 2007; Trondman et al., 2016). All parameters used in REVEALS and all the assumptions of REVEALS are provided in Sugita (2007) and summarised in our paper under methods.

Referee: L254. The plant functional types are already defined as PFT(s) on L90. The authors do not need to repeat it.
Authors. We changed it.

Referee: L258. The authors are better to rewrite the sub-subject title because there may be something wrong with the wording (?)
Authors: We changed it to: „*Methods used to compare DGVM-simulated and REVEALS-estimated tree cover*"

Referee: L268. 'To evaluate ..., the squared chord distance (Prentice, 1980) is calculate for each time window.' I understand what the authors try to say, but it's grammatically incorrect?
Authors: We changed this sentence to: "*The squared chord distance (Prentice, 1980) was calculated for each time window to evaluate the spatial dissimilarities over time between REVEALS, JSBACH and LPJGUESS in terms of total, deciduous, and evergreen tree cover.*"

Results
Referee: Given this study's purpose, I am unsure if Figure 3 is needed in the main text. Figures 4, 5, and 6 alone would be sufficient to characterise the simulated and pollen-based tree-cover estimates over Europe since 8ka. Even if the authors discuss simulation results in areas that cannot be compared to the reconstruction, it is impossible to determine whether they are true or wrong. Descriptions and redundancies regarding regions that are not comparable to the reconstruction may be omitted from the main text as much as possible.
Authors: We see the referee's point, but although the REVEALS dataset used does not cover all of Europe, it is still of interest within this paper to show the differences between the two DGVMs simulations and explain their causes. We therefore kept Figure 3 and the related text.

Referee: L280. 'natural vegetation history' to 'natural vegetation variation' (?)
Authors: We kept „history".

Referee: L286. '... LPJ-GUESS with fewer trees' to '... LPJ-GUESS, with fewer trees'
Authors: done.

Referee: L298. 'in most of the represented regions with highest tree cover fractions' to 'in most regions with the highest tree cover fractions'
Authors: done.

Referee: L337. 'the inter-model spread'
Authors: done.

Referee: L380. 'as driver' to 'as a driver'
Authors: done.

Referee: L400. What is 'This'?
Authors: „This" refers to the definition of the threshold via statistics and not via absolute values. We now write: „*This statistical definition of the threshold allows JSBACH to be still rated in the category "agreement" with larger absolute differences to REVEALS.*"

Discussion

Referee: Is it necessary to thoroughly discuss issues that had already been pointed out before this study began, for example, the 'effect of modern parametrization and tunning of the model to modern conditions' and 'a shortcoming of the REVEALS model and pollen-based plant cover reconstruction'? Shouldn't much description, especially on tree cover reconstructions (assumptions and methodology), be done in the introduction rather than in the discussion because they are known issues and should be the background to this study?
Authors: We think that the detailed and systematic discussion of the potential reasons for the model-data mismatch is the strength of the paper. We described all models, their assumptions and parameter settings in Methods. We then discuss which of these assumptions and/or parameter settings and/or other factors may explain the mismatches between DGVMs and between DGVMs and REVEALS plant cover. We agree that we had not separated this clearly in the REVEALS part. Therefore, we have now deleted some parts of the text from the discussion and added them in a shorter version to the REVEALS methods section. See response above for details on the same issue.

Referee: L407. 'the model agreement to' to 'the model agreement with'
Authors: done.

Referee: L407. Remove 'that is different' (?)
Authors: We kept it.

Referee: L410. Remove 'the in'
Authors: Thank you. We removed it.

Referee: L429. 'the Appendix B' to 'Appendix B'
Authors: done.

Referee: L429. 'we compare basically the climate that is prescribed ...' to 'we compare the climate prescribed …'
Authors: Thank you, the short introduction to the climate comparison was indeed misleading. We now write:
"*Based on a previous study (*Dallmeyer et al., 2021*) analysing a slightly different Holocene MPI-ESM1.2 simulation, we assume summer temperature to be the main climatic driver of the simulated vegetation in the European regions considered here. To infer possible biases in the simulated climate trend, we compare the LPJ-GUESS temperature forcing data with chironomid-based reconstructions of the temperature of the warmest month (Twarm) (Fig. 7), extracted from the synthesis of *Kaufman et al. (2020)* and few other sources (references are given in Table B1 in Appendix B).*"

Referee: L430. 'We assume that summer temperature is the main climatic driver of the vegetation in the regions considered here, ...' The assumption and its analysis are a vital part of the study. They should be described in the methodology section (Section 2), and the analysis results should be written in the result section (Secton 3). Only real arguments should be written here (Secton 4).
Authors: We are not sure we understand the referee's concern. The study is largely based on a transient simulation carried out in an Earth system model. This model is a fully coupled atmosphere-ocean-land-surface model. The vegetation cover is calculated dynamically in the model, in response to climatic (and ecological) changes. However, from a previous study, we know that in Europe, the changes in vegetation in the model MPI-ESM during the simulation are mostly influenced by changes in the temperature of the warmest month (Twarm). Therefore, in this study we focus on the comparison of changes in Twarm between the simulation and pollen-independent reconstructions. It is not a central part of our vegetation comparison. The aim of comparing the climate is rather to exclude that climate biases in the model are responsible for the differences in the vegetation between reconstructions and the models. Therefore, we didn't change the general structure of the paper, but we have rephrased the short introduction into the climate comparison (see answer to the previous comment).

Referee: L437. 'in the temperate and subarctic regions'
Authors: done.

Referee: L507. 'there are various technical reasons that can lead ...' to 'various technical reasons can lead …'
Authors: Thank you, done.

Referee: L515. Remove 'single'
Authors: correction done.

Referee: L522. 'the establishment'
Authors: done.

Referee: L537. 'the disregard of wetlands' to 'the disregard for wetlands'
Authors: done.

Referee: L538. What is 'This'?
Authors: „This refers to the overestimation of forest cover in LPJ-GUESS". Corrected accordingly to "*This potential overestimation….*"

Referee: L541. About this subject, the authors introduce the following sentence in the introduction; 'the DGVM parametrization (bioclimatic limits, disturbance intervals, fire regimes, etc.) are commonly static and based on the current state of land cover although it is characterized by unstable vegetation composition due to rapidly changing natural and anthropogenic stressors (Hengl et al., 2018).' Is this a known issue? If this is a known problem, shouldn't this sensitivity experiment also be described first in Sections 2 and 3, not here?
Authors: It is a known problem that model parameterizations are static, adapted to modern conditions and used to tune the models. This is how models are built-up. But so far – at least to our knowledge – no one has analysed it specifically for a concrete palaeo-climate context. We would like to keep it in the discussion section as we are discussing the potential reasons for the mismatches between the reconstructions and the model results. Inappropriate model parameters are only one possible explanation, that cannot really be evaluated. We do not know which model parameters would represent past climate and vegetation characteristics best.

Referee: L578. Basic information on European land cover change during the Holocene should be presented in the introduction first, as it is part of the background to this study.
Authors: We see this point. However, we think that this information also fits well into the discussion section about the land-use.

Referee: L579. 'the first traces'
Authors: done.

Referee: L580. '... 10000 years ago an its spread' to '... 10000 years ago and its spread'
Authors: Thank you for carefully reading the manuscript. We corrected it.

Referee: L584. 'insights of' to 'insights into'
Authors: done.

Referee: L585. 'a high (preferably annual) resolution'
Authors: done.

Referee: L615. 'Many of the assumptions of the REVEALS model are violated in the "real world" and/or violated in the past, which has been described and discussed in detail earlier (references).' It is an important background for this study, and the authors should describe it in the earlier section, for example, introduction or method. The Pollen-based tree-cover reconstruction using the REVEALS model is the basis of the data-model comparison in this study.
Authors: We agree and moved part of this section to the method section.

Referee: L621. Remove 'as such'
Authors: We deleted this part.

Referee: L624. Remove 'that have been'
Authors: We deleted this part.

Referee: L626. 'Trondman et al., (2016) showed ... can' to 'Trondman et al. (2016) showed ...could'
Authors: We deleted this part.

Referee: L628. Remove 'do'
Authors: We deleted this part.

Referee: L637. 'advise to use' to 'advise using'
Authors: We deleted this part.

Referee: L648. What is 'This'?
Authors: We added „This bias…"

Referee: L650. 'Beside' to 'Besides' (?)
Authors: We deleted this sentence.

Referee: L654. What is 'This'?
Authors: We added „This shortcoming…"

Referee: L661. What is 'This'?
Authors: We added „This limitation …"

Referee: L668. 'a few caveats of the REVEALS model and dataset used in this study can contribute ...' (?)

Authors: We have changed the sentence to: "In summary, violation of model assumptions and other caveats of the REVEALS model itself on the one hand and the REVEALS dataset used in this study on the other hand need to be considered as possible contribution to the discrepancies between REVEALS-estimated and DGVM-simulated tree cover."

Referee: L713. Remove 'a'

Authors: done.

Figures

Referee: L760. It is not easy to figure out Figure 5. Maybe a table would be easier to understand than the diagram (?)

Authors: Thank you. We have labelled the bars to make this figure easier to understand.

Referee: L765. It is not easy for some people to see the green-orange-red color scale in Figure 6. It may be better to change the colours.

Authors: Thank you, this is a very valuable comment. We changed the colours.